# Plug-and-Play Compositionality for Boosting Continual Learning with Foundation Models

**Weiduo Liao[1,3,4], Fei Han[2], Hisao Ishibuchi[3]\*, Qingfu Zhang[4]\*, Ying Wei[1]** \*

[1]Zhejiang University, [2]The Hong Kong University of Science and Technology
[3]Southern University of Science and Technology, [4]City University of Hong Kong
`liaowd@mail.sustech.edu.cn, fhanac@connect.ust.hk`
`hisao@sustech.edu.cn, qingfu.zhang@cityu.edu.hk`
`ying.wei@zju.edu.cn`

## Abstract

Vision learners often struggle with catastrophic forgetting due to their reliance on class recognition by comparison, rather than understanding classes as compositions of representative concepts. This limitation is prevalent even in state-of-the-art continual learners with foundation models and worsens when current tasks contain few classes. Inspired by the recent success of concept-level understanding in mitigating forgetting, we design a universal framework CompSLOT to guide concept learning across diverse continual learners. Leveraging the progress of object-centric learning in parsing semantically meaningful slots from images, we tackle the challenge of learning slot extraction from ImageNet-pretrained vision transformers by analyzing meaningful concept properties. We further introduce a primitive selection and aggregation mechanism to harness concept-level image understanding. Additionally, we propose a method-agnostic self-supervision approach to distill sample-wise concept-based similarity information into the classifier, reducing reliance on incorrect or partial concepts for classification. Experiments show CompSLOT significantly enhances various continual learners and provides a universal concept-level module for the community[1].

## 1 Introduction

Artificial intelligence systems mimic the learning behavior of human intelligence by collecting information and managing knowledge pools from continually assigned tasks in the open world. This need to handle non-independent and identically distributed training data has driven research in continual learning (CL) (Zhou et al., 2024c;a; Biesialska et al., 2020), which aims to balance the objectives of overcoming *catastrophic forgetting* (McCloskey & Cohen, 1989) of learned tasks and achieving *efficient adaptation* to future tasks, also known as the *stability-plasticity dilemma* (Grossberg, 2012). Leveraging a powerful pre-trained backbone to ensure a basic understanding of the world, CL methods of foundation models (FMs), including prompt-based methods (Gao et al., 2023; Smith et al., 2023; Wang et al., 2022c;b; 2024; Gao et al., 2024), representation-based methods (Zhou et al., 2025; 2024b; McDonnell et al., 2023; Zhang et al., 2023), and model-mixture-based methods (Gao et al., 2023; Wang et al., 2024; Marouf et al., 2024), have emerged as a popular direction in this field. However, FMs need to be updated when encountering out-of-distribution data in the upcoming tasks (Yang et al., 2025).

The human brain exhibits *compositionality* (Hupkes et al., 2020; Liao et al., 2024) when comprehending the world, decomposing seen concrete objects into abstract concepts. For example, a *Chihuahua* consists of general dog concepts such as *body shapes* and chihuahua-specific concepts like *small size* and *head shapes*. This interpretability is intuitive to humans, enabling them to generalize novel dog species by decomposing them into combinations of existing concepts while learning disentangled new concepts to refine the knowledge base, thus, facilitating efficient reuse (Liao et al., 2024). A common strategy for existing CL methods for FMs to alleviate forgetting is to inherit parameters learned from

---

\*Corresponding Authors: Ying Wei, Qingfu Zhang, Hisao Ishibuchi

[1]Code is available at github.com/liaoweiduo/CompSLOT.

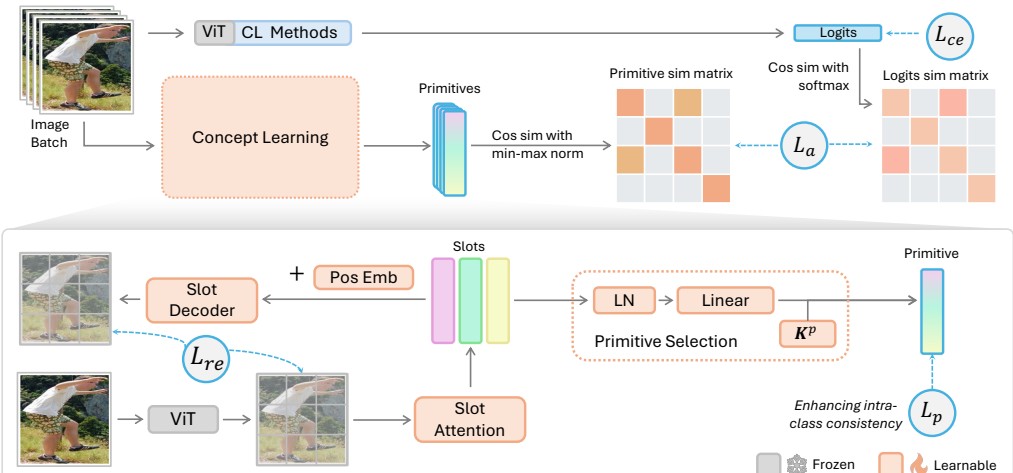

Figure 1: Proposed CompSLOT framework. Given an image batch, we extract primitives for each image with a concept learning procedure. Then, we distill the sample-wise similarity from the primitive representations of the image batch into logits. **Takeaway**: This conceptual pair-wise similarity enables the model to make decisions by additionally considering low-dimensional concept combinations, rather than relying solely on high-dimensional features.

old tasks when initializing new tasks' models, as done in Wang et al. (2024); Gao et al. (2024). These state-of-the-art (SOTA) approaches generally do not fully exploit cross-task potential correlations (i.e., common concepts shared across tasks). In contrast, learning low-dimensional concept combinations to understand classes does not require establishing class representations from the high-dimensional feature level, as in traditional methods, thereby mitigating catastrophic forgetting and enabling rapid adaptation to novel classes (Liao et al., 2024; Yu et al., 2025; Yang et al., 2024; Kundargi et al., 2025; Lai et al., 2024). Thus, a set of CL methods leverages interpretable tools, e.g., ChatGPT (Brown et al., 2020), prototypes (Rymarczyk et al., 2023; Ahrens et al., 2023), and concept bottleneck models (Yu et al., 2025; Yang et al., 2024; Lai et al., 2024) to bring attention to concepts within images, which achieves great success on boosting CL performance. Another challenge is that canonical benchmarks, such as Split-CIFAR100 (Krizhevsky et al., 2009) and Split-ImageNet-R (Hendrycks et al., 2021), are not specifically designed to evaluate the compositionality of continual models. The CFST evaluation framework (Liao et al., 2024) (including CGQA and COBJ) is the only work, to our knowledge, that systematically studies the compositionality of a continual learner. CFST introduces two component-relevant phases in which the data share a common concept set with different combinations. In the first phase, the dataset is split into several continual tasks, aiming to train a continual learner. Subsequently, the second phase is used to evaluate the learner's compositional generalization performance on unseen concept combinations.

Motivated by the above analysis, we pose the research question: *Can the compositionality in concept learning truly enhance the CL performances of SOTA continual learners with FMs?* We propose a **Comp**ositional **Slot** plug-in (**CompSLOT**) for continual learning to answer the above question, which is illustrated in Figure 1. The first step involves extracting concepts from raw images, namely, **conducting concept learning**. Several studies have shown significant progress in concept learning by utilizing explicit concept-level supervision obtained from segmentation masks (Kirillov et al., 2023; Ravi et al., 2025) or natural language annotations (Ramesh et al., 2021; Yu et al., 2025). Nevertheless, it is crucial to compare with SOTA CL methods of FMs, where only labels from the current CL tasks are available as supervision. Consequently, Slot Attention (Locatello et al., 2020), as an SOTA unsupervised object-centric learning approach (Greff et al., 2020), has effectively emerged as a viable self-supervised solution. A Slot Attention module learns to group and encode spatial features into a set of low-dimensional distinct slots, with each slot representing a disentangled region and binding to an object (i.e., concept) in the image. To avoid additional learning of the encoder , the input to Slot Attention can be specified as semantic patch features provided by a pre-trained vision transformer (ViT), which also serves as the learner's backbone. We present a preliminary experiment demonstrating that the learned Slot Attention module exhibits almost no forgetting across compositional tasks, as shown in Figure 2.

With the above method to extract the hidden concepts in images, the next step is to introduce concept learning into CL methods with FMs. The challenge is that there is no unified forwarding framework to organize all CL methods with FMs so that we can easily perform concept learning and assist vanilla learning processes of feature extractors. Hence, we propose regularizing the outputs of learners with **sample-wise similarity based on concepts**. This makes our approach a method-agnostic plugin for any CL method with FM. We first use a learnable aggregation mechanism based on attention to extract class-relevant concepts (i.e., **primitives** (Zou et al., 2024)) as the weighted sum of slots based on their similarity to a learnable task key. The distance of primitives between two images carries information about the similarity in concept level. For example, a *Chihuahua* is close to other dog species (e.g., *German Shepherd*) rather than cat species (e.g., *Siamese*) because they share considerably more concepts (e.g., *dog body*). Subsequently, we propose a method-agnostic primitive-logit alignment plugin to distill the learned sample-wise concept-level similarity into the outputs of models based on a contrastive loss. Our experiments demonstrate that the above procedures successfully select meaningful concepts in images as primitives and ultimately achieve a superior continual learning performance attributed to a better compositional generalization performance.

The contributions of this work are summarized as follows:

- We proposed **CompSLOT**, a method-agnostic plug-in comprising 1) a **concept learning module** that leverages Slot Attention and rich vision foundation models to extract *primitives*, and 2) a **concept knowledge distillation module** that enables learners to intentionally discover *shared and distinct concepts* among classes, thereby guiding the decision-making process of classifiers.
- We designed 1) a **primitive selection mechanism** with an additional **primitive loss** that effectively achieves *robust primitive extraction* across different examples of the same class, and 2) a **primitive-logit alignment loss** that *contrastively regularizes* the *sample-wise similarities* between continual learners' outputs.
- The experimental results demonstrate that CompSLOT successfully leverages **concept-wise compositionality** to significantly enhance **a wide range** of continual learners.

## 2 RELATED WORKS

**Continual Learning of Foundation Models** Benefiting from the rich knowledge in large-scale pre-trained ViT, CL methods with FMs (Zhou et al., 2024a) greatly mitigate forgetting previously learned classification tasks and achieve fast adaptation to new ones. The community has mainly developed three families of approaches, according to the way of utilizing the pre-trained knowledge: 1) *Prompt-based methods* (Gao et al., 2023; Smith et al., 2023; Wang et al., 2022c;b; Gao et al., 2024; Liang & Li, 2024; Le et al., 2024) efficiently tune prompts for tasks rather than fine-tune the backbone; 2) *Representation-based methods* (Zhou et al., 2025; 2024b; McDonnell et al., 2023; Zhang et al., 2023) involve leveraging the advantages of representations from the pre-trained backbone with a class prototype-based classifier; 3) *Model-mixture-based methods* (Gao et al., 2023; Wang et al., 2024; Marouf et al., 2024) utilize hybrid techniques such as model fusion (Wang et al., 2024; Marouf et al., 2024) and model ensemble (Gao et al., 2023) to query a set of models, thus, making the prediction more robust; Moreover, *rehearsing old samples* is an effective way to alleviate forgetting old tasks. Several methods (Wang et al., 2022a; Yan et al., 2021; Zhou et al., 2023) contribute to efficient sample storage mechanisms and auxiliary supervision to address class imbalance, achieving a better stability-plasticity trade-off. However, the above methods ignore hidden conceptual relationships among classes, limiting their significance on handling compositionally relevant tasks.

**Compositionality** Compositionality has been extensively studied in natural language processing (Biesialska et al., 2020; Kaushik & Martin, 2020; Lake & Baroni, 2018; Keysers et al., 2020). To achieve a compositional learner, methods include the introduction of sparse coding (Murphy et al., 2012), regularization (Sun et al., 2016; Luo et al., 2015), and applying independent component analysis (Musil & Mareček, 2022; Yamagiwa et al., 2023). In Hupkes et al. (2020), the authors summarize five types of tests for language compositionality, which are further extended to vision in Liao et al. (2024). Meanwhile, researchers in vision utilize compositional information between objects and attributes to boost zero-shot inference through regularization (Nagarajan & Grauman, 2018), separate learning (Ruis et al., 2021), causal reasoning (Atzmon et al., 2020), self-attention (Khan et al., 2023), and uniting energy-based modules (Wu et al., 2022). Common strategies to learn hidden concepts among continual tasks are external interpretability tools (Yang et al., 2024), learnable

mapping (Lai et al., 2024), prototypes (Rymarczyk et al., 2023; Ahrens et al., 2023; Rymarczyk et al., 2021), CLIP (Kundargi et al., 2025) (Agrawal et al., 2025), ChatGPT (Yu et al., 2025), and assigning different module paths for tasks (Rajasegaran et al., 2019; Ostapenko et al., 2021). Our work, instead, leveraging Slot Attention, does not require prior concept-level supervision for training or an extra concept bottleneck model (Yu et al., 2025), making it more adaptable and easier to integrate with different methods.

**Object-centric Learning** We adopt object-centric learning to autonomously extract concept information directly from images. The introduction of Slot Attention (Locatello et al., 2020) marked the emergence of a new paradigm for disentangling objects (i.e., concepts) within a scene. Subsequent research has focused on improving its robustness in complex environments—primarily through encoder enhancements like covariance regularization (Stange et al., 2023) and bi-level optimization (Jia et al., 2023; Chang et al., 2022). Other efforts have explored advanced decoders to refine decomposition. For example, SLATE (Singh et al., 2022) uses an autoregressive transformer decoder, while Wu et al. (2023); Jiang et al. (2023) propose diffusion-based approaches. Kakogeorgiou et al. (2024) leverages distillation to refine object segmentation via decoder-guided encoder training, and Kori et al. (2023) introduces conditional Slot Attention with a foundational slot dictionary to address specialization limitations. Our method, instead, employs a lightweight MLP decoder to minimize computational cost while preserving effectiveness. Experiments show that this simple design can still significantly benefit continual learning.

# 3 PRELIMINARIES

**Class-incremental vision continual classification tasks** We consider $T$ sequential vision classification tasks with a dataset $\mathcal{D} = [\mathcal{D}^1, \ldots, \mathcal{D}^T]$, where each $\mathcal{D}^t$ consists of image samples $\boldsymbol{x} \in \mathcal{X}^t$ with corresponding labels $y \in \mathcal{Y}^t$. Here, $\mathcal{Y}^t$ is a subset of the global label set $\mathcal{Y}$, and $\forall \mathcal{Y}^t \cap \mathcal{Y}^k = \emptyset$ for $t \neq k$, with task identity unknown during inference, i.e., class-incremental learning (CIL) setting. A general model-based continual learner includes a Vision Transformer (ViT)-based backbone $f(\cdot | \theta_f)$ and classification heads $h_t(\cdot | \theta_{h_t})$, where $t$ is the task identity. Each head is trained separately for the corresponding task, but the outputs from all heads are concatenated for final inference: $\boldsymbol{H_{te}} = f(\boldsymbol{x_{te}} | \theta_f)[0]$, where $[0]$ indicates the [CLS] token (i.e., the first dimension of the output of $f$), and $\mathrm{pred}(\boldsymbol{x_{te}}) = \arg \min [h_1 (\boldsymbol{H_{te}} | \theta_{h_1}) ; \ldots ; h_T (\boldsymbol{H_{te}} | \theta_{h_T})]$, where $[\cdot ; \cdot]$ denotes concatenation.

**Slot attention (Locatello et al., 2020)** As the state-of-the-art object-centric plug-in, slot attention aims to decompose a single image into a set of $K$ disentangled slots $\boldsymbol{S} \in \mathbb{R}^{K \times D_s}$, each encoding one compositional component of the image. $D_s$ is the dimension of slot representation. The output $f(\boldsymbol{x} | \theta_f)$ from a pre-trained ViT backbone consists of two parts: the uninstructed image feature $\boldsymbol{H} = f(\boldsymbol{x} | \theta_f)[0] \in \mathbb{R}^D$ with the token [CLS] and the semantic patch features $\boldsymbol{E} = f(\boldsymbol{x} | \theta_f)[1 :] \in \mathbb{R}^{N \times D}$, where $N$ is the patch number. These $N$ patches are further encoded into the slot space and refined into $K$ slots through an iterative attention procedure. The $K$ slots are first initialized with a learnable Gaussian distribution. In each refinement iteration, slots collect soft assignment information from each patch with an attention mask $\boldsymbol{A} \in \mathbb{R}_+^{K \times N}$. The weighted mean $A$ is then computed along the patch dimension, and a Gated Recurrent Unit (GRU) (Cho et al., 2014) aggregates the patch information into the assigned slots, as follows: $\boldsymbol{A} = \sigma \left( \frac{q(\boldsymbol{S}) k(\boldsymbol{E})^\top}{\sqrt{D_s}} \right), A_{i,n} \leftarrow \frac{A_{i,n}}{\sum_{j=1}^N A_{i,j}}, \boldsymbol{S} \leftarrow \mathrm{GRU}(\boldsymbol{S}, \boldsymbol{A} v(\boldsymbol{E}))$, where $q(\cdot), k(\cdot), v(\cdot)$ are learnable query, key, value projections, respectively, and $\sigma(\cdot)$ is the softmax function.

# 4 METHODS

We present our CompSLOT framework in Figure 1. For each continual task $\mathcal{D}^t$, we first perform **concept learning** (detailed in section 4.1) through a **slot decomposition** and a **primitive selection** mechanism, and then distill the pair-wise similarity statistic of the extracted primitives to model outputs (detailed in section 4.2) in a method-agnostic manner.

Unless otherwise stated, the proposed slot attention and primitive selection modules are **globally shared** across tasks, and no parameters except the ViT backbone are frozen. They are initialized at the beginning of the first CL task. In future CL tasks, their architectures (e.g., the number of slots)

remain fixed, while parameters will be fine-tuned throughout all CL tasks. This design prevents parameter explosion and supports long-sequence tasks, as demonstrated in Figure 3b.

## 4.1 CONCEPT LEARNING

Firstly, we define **concepts** as the ground truth slot decomposition of an image. Since slot attention exhibits permutation equivalence w.r.t. the order of the slots (and masks) (Locatello et al., 2020), we regard $\{\mathcal{S}, \mathcal{A}\}$ as the corresponding set representations of $\{\boldsymbol{S}, \boldsymbol{A}\}$, where $\mathcal{S} = \{\boldsymbol{s_i}\}_{i=1}^{K}$ and $\mathcal{A} = \{\boldsymbol{a_i}\}_{i=1}^{K}$, with $\boldsymbol{s_i} \in \mathbb{R}^{D_s}$ and $\boldsymbol{a_i} \in \mathbb{R}_+^{N}$ being the $i$-th row of $\boldsymbol{S}$ and $\boldsymbol{A}$, respectively.

**Definition 1** (Concept & Disentanglement). Let $\boldsymbol{x}$ be an image, then $\{\mathcal{S}, \mathcal{A}\}$ is a *disentangled* decomposition of $\boldsymbol{x}$ (a.k.a., *concepts* and corresponding *attention regions*), if 1) $\boldsymbol{a_i}, \boldsymbol{a_j} \in \mathcal{A}, \boldsymbol{a_i}, \boldsymbol{a_j} \in \mathbb{R}_+^{N}, \boldsymbol{a_i} \perp \boldsymbol{a_j}$,, and 2) $\mathcal{S}$ satisfies $\arg\min_{\mathcal{S}} \sum_{\boldsymbol{s_i}, \boldsymbol{s_j} \in \mathcal{S}} |\mathrm{sim}(\boldsymbol{s_i}, \boldsymbol{s_j})|$, where $\boldsymbol{s_i}, \boldsymbol{s_j} \in \mathbb{R}^{D_s}$, and $|\cdot|$ is the absolute value, $\perp$ is orthogonal symbol, and $\mathrm{sim}(\cdot, \cdot)$ is a similarity score function, e.g., cosine similarity.

*Remark* 1. The examples of concepts are *Chihuahua's head* w.r.t. *Chihuahua* objects in section 1 and *buildings* w.r.t. images in Figure 2. In Figure 2, $\mathcal{A}$ corresponds to patch regions and $\mathcal{S}$ corresponds to slot representations used to calculate cosine similarity.

To train the slot attention and primitive selection modules, we resort to continually reconstructing $\mathcal{D}$ and an additional contrastive primitive loss.

**Continual image reconstruction** For clarity, we denote the forward path of slot attention as $\{\boldsymbol{S}, \boldsymbol{A}\} = s(\boldsymbol{E}|\theta_s), \boldsymbol{S} \in \mathbb{R}^{K \times D_s}, \boldsymbol{A} \in \mathbb{R}^{K \times N}$. We augment the position embedding into slots $\boldsymbol{S}$ when reconstructing the image, as the ViT does. $\boldsymbol{S'_n} = \boldsymbol{S} \oplus \boldsymbol{pos_n}$, where $\boldsymbol{pos_n} \in \mathbb{R}^{D_s}$ is the learnable position embedding at patch $n$ and $\oplus$ is the element-wise addition with broadcasting. Next, $\boldsymbol{S'} \in \mathbb{R}^{N \times K \times D_s}$, a collection of $N$ position-augmented slots, are mapped back individually to the $D$-dim patch space with an MLP slot decoder $d(\cdot|\theta_d)$. Subsequently, we apply weighted-sum with the attention mask $\boldsymbol{A}$ and finally get the reconstructed patch features $\tilde{\boldsymbol{E}}$. The reconstruction loss $L_{re}$ is the MSE loss between the ground truth patch features $\boldsymbol{E}$ and the reconstructed $\tilde{\boldsymbol{E}}$, as follows:

$$\tilde{\boldsymbol{E}} = \boldsymbol{A}^\top d(\boldsymbol{S'}|\theta_d) \in \mathbb{R}^{N \times D}, \quad L_{re} = ||\boldsymbol{E} - \tilde{\boldsymbol{E}}||_2. \tag{1}$$

We present a preliminary experiment demonstrating that the learned Slot Attention module exhibits almost no forgetting across compositional tasks, as shown in Figure 2. Specifically, we train a Slot Attention module on COBJ 3-tasks as a continual reconstruction task (i.e., trained with reconstruction loss). We then extract slots for images from the first task using the modules after training on the first, second, and third tasks. We observe that each corresponding slot consistently represents a human-interpretable concept and remains stable after training on new tasks, maintaining a high cosine similarity.

**Primitives** When describing the object *Chihuahua*, some concepts (like *Chihuahua's head*) are class-relevant, while others (like *sky*) are class-irrelevant. We name such class-relevant concepts as **primitives**, containing information to identify the desired classes.

**Definition 2** (Primitives). Let $\mathcal{X}^y, \mathcal{S}^y$ be an image set labeled $y$ and the corresponding set of concept sets, respectively, and $\mathcal{S} \in \mathcal{S}^y$, then a concept subset $\mathcal{P} \subset \mathcal{S}$ is *primitives* of $\mathcal{S}$, if $\forall \mathcal{S}' \in \mathcal{S}^y, \mathcal{P} \subset \mathcal{S}'$.

Our goal is to **identify a unified primitive representation** $\boldsymbol{s^p}$, which is regarded as the linear combination of concepts $\boldsymbol{S}$. The basic idea is that primitives have a higher probability appearing in $\boldsymbol{x}$s from the same class $\mathcal{X}^y$ and are likely to carry important information describing this class. Thus, we have the following two questions: 1) *How to represent the selected primitives $\boldsymbol{s^p}$ from $\boldsymbol{S}$?* and 2) *How to minimize the distances among $\boldsymbol{s^p}$s extracted from the images in the same class?*

**Primitive selection** To answer the first question, we propose a learnable attention-based primitive selection mechanism to aggregate $K$ slots. We use a linear module with layer norm and a tanh activation layer to map slots into a unified similarity space. The similarity to a learnable primitive key $\boldsymbol{K^p} \in \mathbb{R}^{D_s}$ measures the slot significance. Then this similarity $\boldsymbol{w_p}$ weights the mapped slots and aggregates them into a single representation $\boldsymbol{s^p}$ (i.e., primitive representation), which is summarized as follows:

$$\bar{\boldsymbol{S}} = \tanh(\mathrm{Linear}(\mathrm{LN}(\boldsymbol{S}))), \quad \boldsymbol{w_p} = \sigma(\tau_t \bar{\boldsymbol{S}} \boldsymbol{K^p}), \quad \boldsymbol{s^p} = \boldsymbol{w_p}^\top \bar{\boldsymbol{S}}, \tag{2}$$

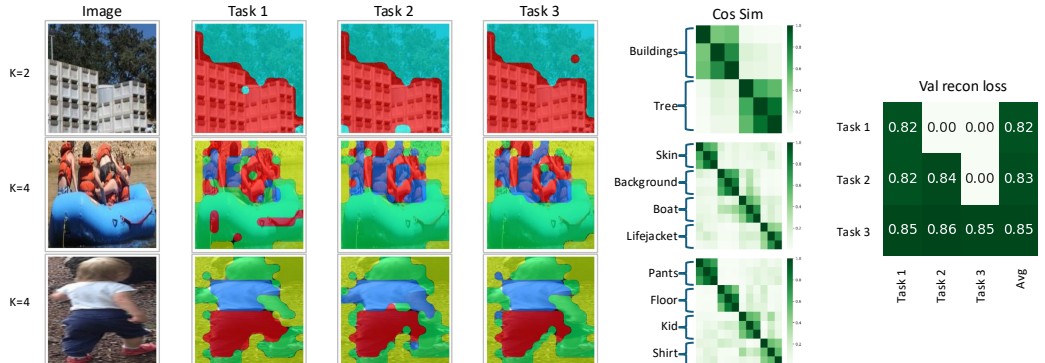

Figure 2: Examples of learned slots by continual COBJ reconstruction tasks and validation reconstruction losses. **Left**: slots are extracted for three example images from the first task on learners after training on the 1-st, 2-nd, and 3-rd tasks. Slots are masked with different colors. **Middle**: corresponding slot cosine similarity matrices grouped by correlated regions. Each group contains slots from three tasks and is identified by the Hungarian matching algorithm. **Right**: validation reconstruction loss matrix. Each row indicates a learner trained after a specific task and evaluated on all seen tasks, respectively. **Takeaway**: learned Slot Attention module enjoys almost no forgetting across compositional-relevant tasks.

where $\tau_t$ is a temperature coefficient controlling the sparsity of slot selection $\boldsymbol{w_p}$, which is set to $100/\sqrt{D_s}$ in practice. A larger $\tau_t$ indicates a smaller number of slots to be selected to represent this image $\boldsymbol{x}$.

**Contrastive primitive loss**   To answer the second question, we rewrite Definition 2 as follows:

**Theorem 1** (Intra-class consistency). *Consider $\mathcal{S}_1, \mathcal{S}_2 \in \mathcal{S}^y$ and two corresponding **largest** primitive sets $\mathcal{P}_1 \subset \mathcal{S}_1, \mathcal{P}_2 \subset \mathcal{S}_2$ are identical, i.e., $\mathcal{P}_1 = \mathcal{P}_2$ and $||\mathcal{P}_1|| = M$, where $||\cdot||$ is the cardinality of set. In other word, consider the pair-wise ordered sets $\{\mathcal{P}_1^\circ, \mathcal{P}_2^\circ\} = \text{match}(\mathcal{P}_1, \mathcal{P}_2)$, where $\text{match}(\cdot, \cdot)$ is a matching algorithm (without loss of generality, Hungarian algorithm (Kuhn, 1955)), then the corresponding matched concepts should be the same: $\mathcal{P}_1^\circ = \{\boldsymbol{s_i^1}\}_{i=1}^M, \mathcal{P}_2^\circ = \{\boldsymbol{s_i^2}\}_{i=1}^M$ and $\forall i \in \{1, \ldots, M\}, \text{sim}(\boldsymbol{s_i^1}, \boldsymbol{s_i^2}) = 1$.*

This form of pair-wise primitive similarities from images within the same class motivates the use of label supervision and contrastive learning (Khosla et al., 2020; Chen et al., 2020b). We first collect the normalized similarity $d_{i,j}^y$ between one-hot label and the softmax similarity $d_{i,j}^s$ between $\boldsymbol{s^p}$. Then, we use a mini-batch clustering loss that a small KL divergence between $d_{i,j}^y$ and $d_{i,j}^s$ means a small distance between $\boldsymbol{s_i^p}, \boldsymbol{s_j^p}$ in the same class and a large distance between those in different classes. The primitive loss $L_p$ is as follows:

$$d_{i,j}^y = \frac{\text{sim}(\mathbb{I}_i, \mathbb{I}_j)}{\sum_{\boldsymbol{x_k} \in B} \text{sim}(\mathbb{I}_i, \mathbb{I}_k)}, \quad d_{i,j}^s = \frac{\exp(\tau_p \text{sim}(\boldsymbol{s_i^p}, \boldsymbol{s_j^p}))}{\sum_{\boldsymbol{x_k} \in B} \exp(\tau_p \text{sim}(\boldsymbol{s_i^p}, \boldsymbol{s_k^p}))}, \quad L_p = \sum_{x_i, x_j \in B} d_{i,j}^y \log \frac{d_{i,j}^y}{d_{i,j}^s},$$
(3)

where $\mathbb{I}_i$ is the one-hot label for sample $\boldsymbol{x_i}$, and $\tau_p$ is a temperature coefficient controlling the strength of primitive loss. The learned slot visualizations in section G (including CGQA, COBJ, ImageNet-R, CIFAR-100) demonstrate that meaningful concepts (represented by primitives, third column "Sum") remain stable across tasks for the same images. We attribute this robustness to "concept rehearsal": although class labels change, many visual concepts are shared and recur across tasks, helping stabilize the primitive selection weights. Section G also visualizes the pair-wise primitive similarities, showing that concept relationships are preserved across images of the same class and shared concepts remain consistent even when images are from different tasks.

By jointly minimizing $L_{re}, L_p$, the learned slot attention module equips the abilities of extracting concepts, identifying primitives, and achieving intra-class primitive consistency. Specifically, we group these losses as $L_{slot} = L_{re} + \alpha L_p$, where $\alpha$ is a coefficient to balance the impact of $L_p$.

## 4.2 Method-agnostic Primitive-Logit Knowledge Distillation

The learned primitive $s^p$ equips a superior property of aggregating important class-relevant concepts. Such understanding can be a self-supervision to regularize the output of the continual learner, i.e., the distribution of logits. Thus, the model gives predictions based on the exact extracted concepts. For example, a *chihuahua* image should have relatively higher logits on other *dog* classes than logits on *cat* classes because they share similar concepts such as *dog body shapes*. Specifically, we design a primitive-logit alignment loss to contrastively distill the learned primitive statistics to logit statistics, i.e., minimizing the KL divergence between softmax logit similarity $d^l$ and previously learned primitive similarity $d^s$, as follows:

$$d^s_{i,j} = \frac{\text{sim}_+(s^p_i, s^p_j)}{\sum_{x_k \in B} \text{sim}_+(s^p_i, s^p_k)}, \quad d^l_{i,j} = \frac{\exp(\tau_a \text{sim}(l_i, l_j))}{\sum_{x_k \in B} \exp(\tau_a \text{sim}(l_i, l_k))}, \quad L_a = \sum_{x_i, x_j \in B} d^s_{i,j} \log \frac{d^s_{i,j}}{d^l_{i,j}},$$

$$(4)$$

where $l_i = h_t(H_i)$ is the logits of $x_i$ for the current task, $\text{sim}_+(\cdot, \cdot)$ is cosine similarity with min-max normalization, and $\tau_a$ is a temperature coefficient controlling the loss strength. We employ min-max normalization (instead of softmax) to sharpen slot supervision. Note that $L_a$ is method-agnostic as long as the CL method has an FM backbone to support extracting semantic features. Finally with the cross-entropy task loss $L_{ce}$, the training loss is as $L_{tr} = L_{ce} + \beta L_a$, where $\beta$ is a coefficient to balance the impact of $L_a$.

## 5 Experiments

In the experiment part, we highlight the research question we will answer: *How and why does our CompSLOT benefit a large range of continual learning with foundation models?* To answer this, we compare algorithms with and without CompSLOT and perform ablation studies in section 5.2. We analyze the influences of hyperparameters in section H.2, investigate different backbones in section H.4 and compare with other concept learning methods in section H.5, and visualize the slot extraction to analyze how CompSLOT enhances CL performance in section G.

### 5.1 Experimental Settings

**Baselines** To verify the universality of the proposed CompSLOT, we adopt a wide range of SOTA continual learners with foundation models, including: 1) **prompt-based methods**: CPrompt (Gao et al., 2024); 2) **representation-based methods**: ADAM+adapter (Zhou et al., 2025), RanPAC (McDonnell et al., 2023), EASE (Zhou et al., 2024b); 3) **Model-mixture-based methods**: CoFiMA (Marouf et al., 2024), FOSTER* (Wang et al., 2022a), DER* (Yan et al., 2021), MEMO* (Zhou et al., 2023). Methods with a "*" postfix indicate that they adopt a rehearsal process. Algorithms are implemented using the PILOT (Sun et al., 2025) platform with default hyperparameters. Methods with CompSLOT are denoted with a postfix "†". Unless otherwise stated, the backbone is ViT-B/16 backbone pretrained on ImageNet-21K, while we also investigate the effect of different backbone architectures in section H.4. We also compare recent concept bottleneck models for continual learning, including CLG-CBM (Yu et al., 2025), and another concept knowledge plugin, SACK (Kundargi et al., 2025) integrated with CODA-Prompt (Smith et al., 2023). In this experiment, we use CLIP ViT-B/16 (Radford et al., 2019) backbone for CompSLOT to extract slots. For the details of the implementation details, please refer to section E.

**Benchmarks** We conduct experiments on compositional datasets, including CGQA and COBJ (Liao et al., 2024), and commonly used datasets, including ImageNet-R (Hendrycks et al., 2021). The former classification datasets contain a sufficient number of combinations of concepts, allowing for visual analysis and evaluating the compositionality. When comparing with other concept-based methods, we conduct experiments on CUB200 (Welinder et al., 2010) and CIFAR100 (Krizhevsky et al., 2009). We choose different continual task settings to evaluate different compositionality levels. Specifically, we denote "**F-S tasks**" as that the first task contains **F** classes and the following tasks contain **S** classes. For example, "50-10 tasks" means splitting 100 classes into six tasks with sequence of class numbers $[50, 10, 10, 10, 10, 10]$. In the main context, we report 10-10 tasks results for CGQA. For results on other benchmarks, please refer to section H.3. All the experiments are conducted on a single Tesla V100 GPU and we analyze the computational cost in section H.8.

Table 1: Main result on CGQA. Methods with CompSLOT are denoted with a postfix "†". Methods rehearse old samples are denoted with a postfix "*". We report results over 3 trials with (mean ± 95% confidence interval). We replace the backbones of all methods to Imagenet-21K-pretrained ViT-B/16.

| Methods | Continual | | | CFST | |
| | AA (%) ↑ | CA (%) ↑ | FF (%) ↓ | Hn (%) ↑ | R↑ |
|---|---|---|---|---|---|
| CPrompt | 46.753±0.570 | 60.179±1.695 | **15.670±0.950** | 78.063±0.817 | 0.964 |
| CPrompt † | **48.537±0.427** | **61.483±1.645** | 18.315±1.111 | **79.091±1.086** | **0.969** |
| ADAM + adapter | 41.930±1.141 | 53.983±0.444 | 13.800±0.187 | 68.649±0.259 | 0.932 |
| ADAM + adapter † | **49.480±1.201** | **60.989±0.641** | **12.896±0.379** | **74.335±0.572** | **0.958** |
| RanPAC | 65.810±0.802 | 75.504±0.318 | 10.515±0.176 | 78.868±0.918 | 1.016 |
| RanPAC † | **66.753±0.867** | **76.584±0.603** | **10.219±0.281** | **79.815±0.829** | **1.032** |
| EASE | 47.657±1.494 | 59.475±2.574 | **18.215±0.107** | 79.713±0.449 | 0.996 |
| EASE † | **49.323±1.165** | **62.603±1.252** | 22.470±2.472 | **82.887±0.320** | **1.001** |
| CoFiMA | 65.107±0.508 | 73.227±1.047 | 15.248±0.542 | 86.711±0.483 | 1.011 |
| CoFiMA † | **66.170±0.578** | **74.322±0.463** | **14.204±0.880** | **88.297±0.278** | **1.017** |
| FOSTER* | 60.863±0.271 | 68.800±0.496 | **2.441±0.122** | 89.791±0.086 | 1.087 |
| FOSTER* † | **66.290±1.451** | **71.828±2.619** | 6.470±5.770 | **89.910±0.710** | **1.154** |
| DER* | 52.003±1.019 | 62.675±1.695 | 40.122±0.907 | **90.119±0.510** | 1.080 |
| DER* † | **54.900±1.093** | **66.020±1.049** | **38.941±0.995** | 88.986±0.129 | **1.096** |
| MEMO* | 56.553±1.804 | 66.462±0.702 | 9.289±0.326 | 82.425±1.282 | 1.029 |
| MEMO* † | **58.653±1.449** | **68.037±1.459** | **8.944±0.268** | **84.003±1.451** | **1.050** |

**Metrics** For continual training stage, we report the average accuracy of all tasks after training the last task $\mathbf{AA} = \frac{1}{T} \sum_{t=1}^{T} \mathbb{E}_{(x_{te},y) \in \mathcal{D}_{te}^t} [\Delta(\text{pred}(x_{te}|P_T), y)]$, the average cumulative accuracy for each task $\mathbf{CA} = \frac{1}{T} \sum_{t=1}^{T} \frac{1}{T-t+1} \sum_{u=t}^{T} \mathbb{E}_{(x_{te},y) \in \mathcal{D}_{te}^u} [\Delta(\text{pred}(x_{te}|P_t), y)]$, and average forgetting for each task $\mathbf{FF} = \frac{1}{T} \sum_{t=1}^{T} \mathbb{E}_{(x_{te},y) \in \mathcal{D}_{te}^t} [\Delta(\text{pred}(x_{te}|P_t), y)] - \text{AA}$, where $\mathcal{D}_{te}^t$ is the testing dataset for task $t$ and $\Delta(\cdot, \cdot)$ is the equal function. After training on all continual tasks, specifically for CGQA and COBJ, we perform CFST on five compositional test suites including **sys, pro, sub, non, noc**, which contain *novel recombinations, more combinations, shifting attributes, seen combinations, novel concepts* of testing samples, respectively. We generate 300 few-shot tasks for each test suite. For clarity, we calculate the Harmonic mean (i.e., $\mathbf{Hn} = 3/(1/\text{sys} + 1/\text{pro} + 1/\text{sub}), \mathbf{Hr} = 2/(1/\text{non} + 1/\text{noc})$, as suggested in Liao et al. (2024). Then we report Hn and the ratio of Hn and Hr (i.e., $\mathbf{R} = \text{Hn}/\text{Hr}$). For detailed results on each compositional test suite, please refer to section H.1. Larger Hn and R indicate that the extracted features have better compositional generalization performance.

## 5.2 RESULTS

**Overall results** We report the statistical results in Table 1. Across all baselines, CompSLOT consistently enhances performance, with the most significant improvement observed in ADAM+adapter (absolute gain: +7.550 in AA). Notably, CA and FF demonstrate consistent superiority over other methods (except CPrompt and FOSTER, because the original methods do not perform well on the finished tasks, thus, forget less), indicates that our CompSLOT not only mitigates catastrophic forgetting of old tasks but also preserves strong forward adaptation to novel tasks. This robustness is primarily attributed to CompSLOT's improved compositional generalization (manifested by higher Hn and R scores), confirming its ability to learn latent conceptual units and dynamically compose them for robust classification across diverse methodological frameworks.

**Learning curve** Figure 3a shows the learning curves of all methods on the 10-10 tasks from CGQA. We observe that concept learning significantly improves continual learning performance across the entire training process, demonstrating its ability to stabilize learning and mitigate forgetting.

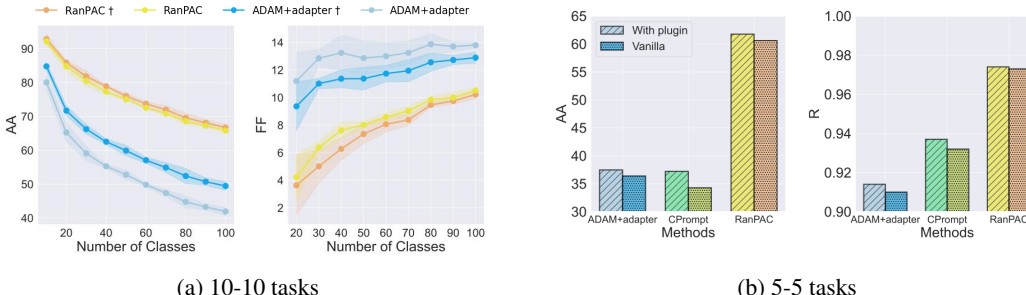

(a) 10-10 tasks            (b) 5-5 tasks

Figure 3: Learning curves and histograms of methods with and without CompSLOT on CGQA a) 10-10 tasks and b) 5-5 tasks. Slot is the case directly using the primitive representation and a cosine similarity classifier for the continual tasks.

**Long task sequence** Figure 3b presents the comparative performance analysis across a challenging long-task sequence of 5-5 CGQA tasks. The results reveal that CompSLOT consistently enhances both continual learning accuracy and compositional generalization performance, even when the slot attention module globally shared across all continual tasks. This finding underscores the remarkable robustness of employing slot attention mechanisms for boosting concept learning in CL scenarios. Notably, the stable improvement suggests that CompSLOT effectively captures transferable compositional knowledge, enabling better adaptation across sequential tasks without task-specific customization.

**Ablation studies** To evaluate the contribution of each proposed component, we conduct comprehensive ablation experiments, with results presented in Table 2. First, to rule out the possibility that performance gains stem solely from **increased model capacity**, we expand the hidden representation dimensions (denoted as "+param") in RanPAC and CPrompt (see section E for details) to match the parameter count of RanPAC † and CPrompt †, respectively. We further perform the following controlled experiments: 1) **Primitive loss ablation** ($L_p$): We remove the primitive loss term and replace the primitive selection mechanism with a simple slot averaging strategy (avg). 2) **Slot-selection function ablation**: We substitute the softmax operation in Equation 2 with alternative weighting methods, including: averaging (avg), sigmoid (sig), sign quantization (sign), and cosine similarity (cos). Across both methods, disabling

Table 2: Ablation results on CGQA.

| Methods | $L_p$ | $L_a$ | AA (%) ↑ | R↑ |
|---|---|---|---|---|
| RanPAC | ✗+param | ✗ | 65.080 | 1.010 |
|  | ✗avg | ✓ | 58.220 | 0.969 |
|  | ✓avg | ✓ | 65.870 | 1.003 |
|  | ✓sig | ✓ | 65.950 | 1.020 |
|  | ✓sign | ✓ | 65.140 | 1.006 |
|  | ✓cos | ✓ | 63.910 | 0.989 |
|  | **✓soft** | **✓** | **66.753** | **1.032** |
| CPrompt | ✗+param | ✗ | 46.300 | **0.969** |
|  | ✗avg | ✓ | 40.230 | 0.952 |
|  | ✓avg | ✓ | 47.690 | 0.958 |
|  | ✓sig | ✓ | 48.080 | 0.961 |
|  | ✓sign | ✓ | 47.780 | 0.966 |
|  | ✓cos | ✓ | 47.410 | 0.964 |
|  | **✓soft** | **✓** | **48.537** | **0.969** |

$L_p$ or altering the slot-selection mechanism leads to significant degradations in AA and R scores, demonstrating the critical importance of each component. 1) The primitive loss $L_p$ ensures intra-class consistency, which is vital for reliable primitive selection and, consequently, improved concept-level class understanding. On the other hand, using all slots indiscriminately allows less relevant concepts (e.g., background) to dilute class-relevant ones, leading to confusion. 2) The softmax-based weighting (as formulated in Equation 2) provides a selection with a convex combination of slots in one image to ensure the primitive representations of images are within an appropriate range, which makes the training robust. A more comprehensive ablation study can be found in section H.6.

**Concept learning and visualization** To evaluate whether the learned concepts align with the ground truth, we establish evaluation experiments and design six metrics including the slot representation MAE and the slot mask mIOU. Due to page limit, we describe the details of the metrics and the experiments in section H.2. After that, we visualize the learned concepts and compare primitive similarity with ground truth concept similarity in Figure 4. We observe that CompoSLOT consistently identifies *Other shoes*, *Person*, and *Chair*, which are important concepts (primitives), in an unsuper-

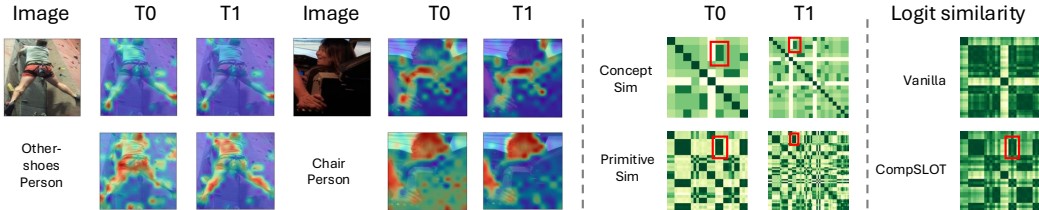

Figure 4: Concept learning visualization and fruits of primitive-logit knowledge distillation on COBJ. **Left**: Examples of visualization of learned slots that associated with the corresponding concepts after finish T0 and T1 tasks. Red color indicates a high value in slot attention masks. **Middle**: Primitive (comparing with ground truth concept) cosine similarity matrix for ADAM+adapter † on 30 images in T0 (left) and on additional 30 images in T1(right). **Right**: Logit cosine similarity for ADAM+adapter with and without CompSLOT on 30 images in T0. **Takeaway**: The learned primitive successfully mimics concept statistics and CompSLOT successfully distills pair-wise primitive similarity into logits. Red box shows evidences.

vised manner. The frequent existence of concepts between different tasks shows a *concept rehearsal* phenomenon: although class labels change, many visual concepts are shared and recur across tasks, helping stabilize the primitive selection. The learned primitives mimic concept statistics in terms of cosine similarity and the proposed primitive-logit alignment loss successfully distills pair-wise primitive similarity into logits, which demonstrate that the models make decisions by additionally considering low-dimensional concept combinations instead of only relying on high-dimensional features. For the details of concept and similarity visualization, please refer to section G.

**Influences of hyperparameters** CompSLOT introduces several hyperparameters mainly in the following three mechanisms: *concept learning stage, slot attention architecture*, and *primitive-logit knowledge distillation stage*. We conduct comprehensive experiments to investigate the influences of each hyperparameter in section H.2. Here we specifically showcase the effect of primitive-logit alignment loss coefficient $\beta$ when continual training CPrompt on the first three tasks of the 10-10 CGQA as an example. The results are shown in Table 6. We observe that AA increases as $\beta$ increases but decrease after a threshold (around 2). This indicates that an excessively large $\beta$ hinders the effectiveness of CPrompt's smooth regularization, leading to conflicts. However, within an appropriate range, our CompSLOT works effectively with CPrompt's smooth regularization.

## 6  CONCLUSION

This work propose **CompSLOT**, a framework introducing **concept learning** into the continual learning paradigm for foundation models. The proposed **primitive selection mechanism** effectively extracts class-relevant concepts while maintaining robustness across extended task sequences. Meanwhile, the **primitive-logit knowledge distillation** mechanism enforces concept-based sample similarity regularization, enabling lightweight adaptation to diverse CL methods with foundation models. Experimental results confirm that the performance improvements stem from enhanced **compositional generalization**, offering a novel **concept-level perspective** for the continual learning community. A limitation of our current approach is that concept learning must precede providing conceptual self-supervision to the CL task. Future work will explore end-to-end integration of our mechanism into the continual learning pipeline and study the joint effect when combining with regularization methods that also manipulating the logits. We hope this research inspires further advancements in developing resilient and interpretable vision models.

## 7  ETHICS STATEMENT

We hereby affirm our strict adherence to the ICLR Code of Ethics. We have carefully considered the ethical implications of our research throughout the entire process of study design, data collection, experimentation, and manuscript preparation, and we confirm that our work does not violate any of the principles outlined in the ICLR Code of Ethics. All datasets used, including ImageNet-R,

CIFAR100, CUB200, CGQA, COBJ, were sourced in compliance with relevant usage guidelines, ensuring no violation of privacy. No personally identifiable information was used, and no experiments were conducted that could raise privacy or security concerns. We are committed to maintaining transparency and integrity throughout the research process.

## 8    REPRODUCIBILITY STATEMENT

We are committed to ensuring the reproducibility of our research presented in this paper. To facilitate the replication of our results and the verification of our findings, we have provided comprehensive implementation details in section E. Additionally, the datasets we used, are publicly available, ensuring consistent and reproducible evaluation results.

### ACKNOWLEDGMENTS

We thank all the anonymous reviewers for their constructive suggestions on improving this paper. This work was supported in part by grants from the National Natural Science Foundation of China (No. NSFC62441236), the Research Grants Council of the Hong Kong Special Administrative Region, China (GRF Project No. CityU11212524), National Natural Science Foundation of China (Grant No. 62376115), and Guangdong Provincial Key Laboratory (Grant No. 2020B121201001).

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

# A  APPENDIX CONTENTS

## CONTENTS

## B   DISCUSSIONS

**Classification Bias from a Concept-Combination Perspective**    The root cause of compromised stability and plasticity often lies in sub-optimal classifier design, particularly when classifiers develop reliance on inaccurate or incomplete feature representations due to concept biases in training data. To illustrate, consider a scenario where the current vision task $T_1$ contains two specific classes (*a human standing by a tree* and *a human inside a boat*) alongside other classes that lack human-related concepts. In such cases, the learned classifier might develop an over-reliance on distinguishing these two classes based solely on *tree* and *boat* concepts while neglecting the more critical *human* attribute. This limited conceptual understanding creates significant generalization problems when encountering unseen concept combinations. For instance, during task $T_2$, a novel image labeled *a pig inside a boat* would likely receive disproportionately high logits for the *human inside a boat* class due to the classifier's inability to properly disentangle object-class relationships from spatial-contextual cues. Conversely, a *human inside a boat* image might similarly activate the *pig inside a boat* class predictions. This conceptual entanglement manifests as catastrophic forgetting in $T_1$ (as evidenced by diminished global accuracies post-training on $T_2$) and severely hampers plasticity for $T_2$ through incorrect plastic responses to novel concept combinations.

**Whether concept sharing is a common phenomenon in the real-world?**    In real-world scenarios, concept sharing is quite common, like, in fine-grained classification cases such as CUB200, and in images with massive objects such as COBJ. This phenomenon is also discussed in other works. For example, Welinder et al. (2010) claims that fine-grained bird classes share some basic parts, and Krause et al. (2015) claims that fine-grained categories share similar shapes. In the experimental results, CompSLOT consistently brings significant improvements to continual learning algorithms on these real-world cases. In contrast, datasets like CIFAR, which have relatively little concept sharing, are uncommon in complicated real-world scenarios.

**Fairness issues**    To demonstrate the effectiveness of our CompSLOT, we list the actions to make the comparison as fair as possible:

1. When comparing with and without CompSLOT, e.g., in Table 1, we used exactly **the same backbone** for both **continual learners and the slot attention**, which was the ViT-B/16 backbone pretrained on ImageNet-21K sourced from the Python timm package. When comparing with other concept-based methods in Table 10, we should note that it is impossible to achieve perfect fairness due to the use of external LLMs in CLG-CBM. We used CLIP ViT-B/16 for CompSLOT to extract slots without changing the backbone of the continual learner to show the method-agnostic nature and the adaptivity of our proposed plugin.
2. When training slot attention, we **DID NOT** introduce additional supervision, such as concept labels.
3. Most of the continual learner-related hyperparameters used their default settings, as suggested in the PILOT platform, while for the additional hyperparameters introduced in this work, please refer to section E.
4. To further ensure fairness and show that performance gains are not from the increased model capacity, we also compared with a case extending the number of parameters in an ablation study in section 5.

## C  ADDITIONAL RELATED WORKS

**Continual learning from scratch**    To mitigate forgetting previously learned vision classification tasks and achieve fast adaptation to new ones, the continual learning community has developed three main families of approaches that do not utilize a pre-trained foundation model:

1. Rehearsal-based methods (Achille et al., 2018; Rolnick et al., 2019; Rahul & Pratik, 2022; Hersche et al., 2022; Sun et al., 2022; Qiu et al., 2023; Sun et al., 2023) store memory-efficient samples or features from past tasks for reviewing knowledge. However, such buffers can cause significant memory overload as the number of tasks increases.
2. Regularization-based methods (Lopez-Paz & Ranzato, 2017; Li & Hoiem, 2017; Kirkpatrick et al., 2017; Achille et al., 2018; Hersche et al., 2022) constrain gradient updates to preserve important knowledge from old tasks, but this can limit the adaptation capability to new tasks.
3. Architecture-based methods (Mallya & Lazebnik, 2018; Douillard et al., 2022; Ring, 1997; Ruvolo & Eaton, 2013; Gaunt et al., 2017; Li et al., 2019; Rajasegaran et al., 2019; Chen et al., 2020a; Mendez & EATON, 2021; Ostapenko et al., 2021; Rahul & Pratik, 2022; Hihn & Braun, 2023) aim to create new modules for upcoming tasks, making the determination of module composition crucial for different tasks.

## D  THEOREM

Firstly, we define *concepts* as the ground truth slot decomposition of an image. Since slot attention exhibits permutation equivalence w.r.t. the order of the slots (and masks) (Locatello et al., 2020), we regard $\{\mathcal{S}, \mathcal{A}\}$ as the corresponding set representations of $\{\boldsymbol{S}, \boldsymbol{A}\}$, where $\mathcal{S} = \{\boldsymbol{s_i}\}_{i=1}^K$ and $\mathcal{A} = \{\boldsymbol{a_i}\}_{i=1}^K$, with $\boldsymbol{s_i} \in \mathbb{R}^{D_s}$ and $\boldsymbol{a_i} \in \mathbb{R}_+^N$ being the $i$-th row of $\boldsymbol{S}$ and $\boldsymbol{A}$, respectively.

**Definition 3** (Concept & Disentanglement, equivalent to Def. 1). Let $\boldsymbol{x}$ be an image, then $\{\mathcal{S}, \mathcal{A}\}$ is a *disentangled* decomposition of $\boldsymbol{x}$ (a.k.a., *concepts* and corresponding *attention regions*), if 1) $\boldsymbol{a_i}, \boldsymbol{a_j} \in \mathcal{A}, \boldsymbol{a_i} \in \mathbb{R}_+^N, \boldsymbol{a_i} \perp \boldsymbol{a_j}$,, and 2) $\mathcal{S}$ satisfies $\arg\min_{\mathcal{S}} \sum_{\boldsymbol{s_i}, \boldsymbol{s_j} \in \mathcal{S}} |\text{sim}(\boldsymbol{s_i}, \boldsymbol{s_j})|$, where $\boldsymbol{s_i} \in \mathbb{R}^{D_s}$, and $|\cdot|$ is the absolute value, $\perp$ is orthogonal symbol, and $\text{sim}(\cdot, \cdot)$ is an arbitrary similarity score function, e.g., cosine similarity.

*Remark* 2 (**Requirement 1**: Disentanglement).  The competitive spatial attention and the limited capability of a slot naturally achieve the orthogonality of $\mathcal{A}$. In practice, $\mathcal{S}$ is encouraged to be orthogonal (slots bind to different concepts in $\boldsymbol{x}$) but not ideal since there are some semantically similar concepts, e.g., *grass* and *leaves*. Such a disentanglement structure is also mentioned in Park et al. (2024); Li et al. (2025).

**Definition 4** (Primitives, equivalent to Def. 2). Let $\mathcal{X}^y, \mathcal{S}^y$ be an image set labeled $y$ and the corresponding set of concept sets, and $\mathcal{S} \in \mathcal{S}^y$, then a concept subset $\mathcal{P} \subset \mathcal{S}$ is *primitives* of $\mathcal{S}$, if $\forall \mathcal{S}' \in \mathcal{S}^y, \mathcal{P} \subset \mathcal{S}'$.

*Remark* 3.  Although $\mathcal{P}$ is defined at the image level, we can also say that it is unambiguously $y$'s primitive set, denoted $\mathcal{P}^y$. In general, $\mathcal{P} \neq \mathcal{S}$, because there are always image-specific concepts in the image, e.g., background.

**Theorem 2** (**Requirement 2**: Intra-class consistency, equivalent to Theorem. 1). *Consider* $\mathcal{S}_1, \mathcal{S}_2 \in \mathcal{S}^y$ *and two corresponding **largest** primitive sets* $\mathcal{P}_1 \subset \mathcal{S}_1, \mathcal{P}_2 \subset \mathcal{S}_2$ *are identical, i.e.,* $\mathcal{P}_1 = \mathcal{P}_2$ *and* $||\mathcal{P}_1|| = M$, *where* $||\cdot||$ *is the cardinality of set. In other word, consider the pair-wise ordered sets* $\{\mathcal{P}_1^{\circ}, \mathcal{P}_2^{\circ}\} = \text{match}(\mathcal{P}_1, \mathcal{P}_2)$, *where* $\text{match}(\cdot, \cdot)$ *is a matching algorithm (without loss of generality, Hungarian algorithm (Kuhn, 1955)), then the corresponding matched concepts should be the same:* $\mathcal{P}_1^{\circ} = \{\boldsymbol{s_i^1}\}_{i=1}^M, \mathcal{P}_2^{\circ} = \{\boldsymbol{s_i^2}\}_{i=1}^M$ *and* $\forall i \in \{1, \ldots, M\}, \text{sim}(\boldsymbol{s_i^1}, \boldsymbol{s_i^2}) = 1$.

**Theorem 3** (**Requirement 3**: Inter-class concept sharing). *If there is a shared primitive subset between* $y_1, y_2$, *all images in* $\mathcal{X}^{y_1}, \mathcal{X}^{y_2}$ *should contain this subset. If* $\exists \mathcal{P} \subset \mathcal{P}^{y_1}, \mathcal{P} \subset \mathcal{P}^{y_2}, ||\mathcal{P}|| = M > 0$, *then* $\forall \mathcal{S}_1 \in \mathcal{S}^{y_1}, \mathcal{S}_2 \in \mathcal{S}^{y_2}, \mathcal{P} \subset \mathcal{S}_1, \mathcal{P} \subset \mathcal{S}_2$.

*Remark* 4 (**Requirement 4**: Inter-task consistency).  After trained on future tasks, the concept sets of the same $\boldsymbol{x}$ should be not changed. In $T$ CL tasks, $\forall \boldsymbol{x} \in \mathcal{X}^u, 1 \leq u < T$, consider the extracted concept sets $\{\mathcal{S}^t\}_{t=u}^T$ after task $t \in \{u, \ldots, T\}$, then $\forall t_1, t_2 \in \{u, \ldots, T\}, \mathcal{S}^{t_1} = \mathcal{S}^{t_2}$.

CompSLOT implicitly encourages **Requirement 1** via slot attention's soft-clustering and supports **Requirements 3–4** empirically (Figure 2) and the visualization experiments in section G. **Require-**

**ment 2** is enforced through a primitive loss, described in Equation 3, ensuring slot stability across class instances.

### D.1 PROOF OF THEOREM 1

*Proof.* By the definition of $\mathcal{P}_1, \mathcal{P}_2$ as the *primitive* sets of $\mathcal{S}_1, \mathcal{S}_2$, respectively, and that $\mathcal{S}_1, \mathcal{S}_2 \in \mathcal{S}^y$, without loss of generality, $\mathcal{P}_2$ is also a primitive set of $\mathcal{S}_1$. Thus, $\mathcal{P}_2 \subset \mathcal{S}_1$. Assume, for the sake of contradiction, that there exists a concept $s$, such that $s \in \mathcal{P}_2$ and $s \notin \mathcal{P}_1$, i.e., $\mathcal{P}_1 \neq \mathcal{P}_2$. Since $\mathcal{P}_1$ is the **largest** *primitive* set of $\mathcal{S}_1$, we must have $\mathcal{P}_2 \subseteq \mathcal{P}_1$ and $\forall \mathcal{P} \subseteq \mathcal{P}_1, s \notin \mathcal{P}$. This contradicts our initial assumption that $s \in \mathcal{P}_2$.

Therefore, the theorem holds.

$\square$

*Remark* 5. The matching algorithm facilitates concept alignment across different sets, thereby enabling the computation of our proposed evaluation metrics in section H.2 as well as supporting the visualizations presented in section G. However, this alignment process introduces significant computational overhead that renders it impractical for integration within our distillation framework. To address this limitation, we propose an attention-based primitive selection mechanism (detailed in section 4.1) that ensures permutation invariance to concept ordering in the extracted primitives, effectively eliminating the need for explicit concept matching. This design choice maintains computational efficiency while preserving the critical semantic relationships required for reliable evaluation and visualization.

### D.2 PROOF OF THEOREM 2

*Proof.* Assume, for the sake of contradiction, that there exists $\mathcal{P}', \mathcal{S}'$ and $\mathcal{P}' \subset \mathcal{P}^{y_1}, \mathcal{P}' \subset \mathcal{P}^{y_2}, ||\mathcal{P}'|| = M > 0, \mathcal{S}' \in \mathcal{S}^{y_1}$ (or $\mathcal{S}^{y_2}$), such that $\mathcal{P}' \not\subset \mathcal{S}'$. By the definition of $\mathcal{P}^{y_1}$ as the primitive set for all $\mathcal{S} \in \mathcal{S}^{y_1}$, thus $\mathcal{P}' \subset \mathcal{S}'$. This contradicts our initial assumption that $\mathcal{P}' \not\subset \mathcal{S}'$.

Therefore, the theorem holds.

$\square$

## E  HYPERPARAMETERS AND EXPERIMENTAL SETTINGS

The hyperparameter settings for the concept learning stage are summarized in Table 3, with key values tuned through validation. For the concept knowledge distillation phase, we maintain fairness in comparison by adopting the platform-default hyperparameters from the PILOT framework for both standard CL baselines and CompSLOT-enhanced variants, with additional parameters introduced in section 4.2 detailed in Table 4. All configurations employ an 80-20 train-validation split using a randomly sampled validation set. To ensure consistent model capacity across methods, all algorithms utilize the ViT-B/16 backbone pretrained on ImageNet-21K as the shared feature extractor unless otherwise stated. When comparing with other concept-based models in Table 10, we use CLIP ViT-B/16 as the CompSLOT's backbone to extract slot, suggested in Yu et al. (2025). The backbone parameters are sourced from the Python timm (Wightman, 2019) package. For the compositional testing in CGQA and COBJ, we used randomly generated 300 few-shot tasks for each test suite, as suggested in Liao et al. (2024). For ablation studies specifically examining CompSLOT's impact, we appropriately scale model capacities through expanded hidden representations: RanPAC: Increased feedforward layer width (ffn_num) from 64 to 256; CPrompt: Extended prompt length (prompt_len) from 50 to 65 tokens. These adjustments ensure fair comparison by matching representational capacity when introducing our architectural modifications, enabling more reliable evaluation of CompSLOT's actual contribution beyond simple capacity increases.

## F  PSEUDO CODE

In the main paper, we propose a two-stage procedure, including concept learning (aiming to extract concept-level representation by performing slot representation training and primitive selection) and

Table 3: Detail hyperparameters for **concept learning stage** in our main experiments.

| Hyper-parameters | Value |
|---|---|
| Optimizer | Adam |
| LR scheduler | Cosine |
| LR (1-st task) | 1e-4 |
| LR (others) | 1e-5 |
| LR (min) | 1e-8 |
| Batch size | 256 |
| Weight decay | 0 |
| Epoch | 50 |
| $D_s$ | 128 |
| $K$ | 10 |
| Slot refinemnt iterations $N_s$ | 5 |
| Slot decoder hidden embedding dim | Linear with ReLU ($128 \rightarrow 256 \rightarrow 256 \rightarrow 768$) |
| $\tau_t$ | 100 |
| $\alpha$ | 10 |
| $\tau_p$ | 10 |

Table 4: Detail hyperparameters for **concept knowledge distillation stage** in our main experiments.

| Methods | $\beta$ | $\tau_a$ |
|---|---|---|
| CPrompt | 10 | 0.05 |
| ADAM + adapter | 10 | 0.5 |
| RanPAC | 15 | 0.5 |
| EASE | 10 | 0.1 |
| CoFiMA | 1 | 0.001 |
| FOSTER | 2 | 0.05 |
| DER | 7 | 0.01 |
| MEMO | 0.05 | 0.1 |

concept knowledge distillation (aiming to distill sample-wise concept-based similarity into logits). We summarize the training framework of CompSLOT in Algorithm 1. Specifically, we perform concept learning in Lines 4-9. The slot attention and primitive selection module are initialized at first. For each batch of samples in task $t$, we perform Algorithm 2 and use the obtained primitive loss and reconstruction loss to train slot attention and primitive selection modules in Line 6. After $E$ epochs of training, we perform concept knowledge distillation in Lines 11-18. We calculate pair-wise primitive similarity and obtain primitive-logit alignment loss with Equation 4 in Line 15. We detail the slot representation learning in Algorithm 2. Specifically, we first obtain semantic patch features in Line 3. Then, we use slot attention module to decompose it into a set of slots in Line 4. Next, we reconstruct the patch feature and obtain the reconstruction loss in Lines 6-8. After that, we calculate the primitives in Lines 10-12 and obtain primitive loss with Equation 3 in Lines 14-15.

---

**Algorithm 1** Continual Learning Framework

---

1: **Input:** # tasks $T$, tasks $\mathcal{D}^1, \ldots, \mathcal{D}^T$, # epochs $E$, candidate CL method $CL(\cdot|\theta_f, \theta_h)$.
2: Initialize slot attention and primitive selection module.
3: **for** $t$ from 1 to $T$ **do**
4:     /* Concept Learning */
5:     **for** $i$ from 1 to $E$ **do**
6:         Sample a batch of images $(\boldsymbol{x}, y) \sim \mathcal{D}^t$.
7:         Perform Algorithm 2 to obtain primitives $s^p$, contrastive primitive loss $L_p$, and reconstruction loss $L_{re}$.
8:         $L_{slot} = L_{re} + \alpha L_p$.
9:         Backward loss and update.
10:     **end for**
11:     /* Concept Knowledge Distillation */
12:     **for** $i$ from 1 to $E$ **do**
13:         Sample a batch of images $(\boldsymbol{x}, y) \sim \mathcal{D}^t$.
14:         Perform Algorithm 2 to obtain primitives $s^p$ without collecting gradients.
15:         Perform CL method to obtain logits $CL(x|\theta_f, \theta_h)$ and task loss $L_{ce}$.
16:         Calculate primitive-logit alignment loss $L_a$.
17:         $L_{tr} = L_{ce} + \beta L_a$.
18:         Backward loss and update.
19:     **end for**
20: **end for**

---

**Algorithm 2** Slot Representation Learning

---

1: **Input:** Image batch $\{x_i\}_{i=1}^B$, CL backbone $\theta_f$, # slots $K$, slot dimension $D_s$, # epochs $E$, .
2: **Output:** Primitive $s^p$, contrastive primitive loss $L_p$, reconstruction loss $L_{re}$.
3: Obtain semantic patch features $\boldsymbol{E} = f(x_i|\theta_f)[1:]$.
4: Obtain a set of $K$ slots and the corresponding attentions $\{\boldsymbol{S}, \boldsymbol{A}\}$.
5: /* Reconstruction Loss */
6: Add position embedding for each patch: $\boldsymbol{S'_n} = \boldsymbol{S} \oplus \boldsymbol{pos_n}$.
7: Decode $\boldsymbol{S'}$ and re-construct using $\boldsymbol{A}$: $\tilde{\boldsymbol{E}} = \boldsymbol{A}^\top d(\boldsymbol{S'}|\theta_d)$.
8: $L_{re} = ||\boldsymbol{E} - \tilde{\boldsymbol{E}}||_2$.
9: /* Primitive Selection */
10: Obtain Mapped slots $\bar{\boldsymbol{S}} = \tanh(\text{Linear}(\text{LN}(\boldsymbol{S})))$.
11: Obtain weights for each slot $\boldsymbol{w_p} = \sigma(\tau_p \bar{\boldsymbol{S}} \boldsymbol{K^p})$.
12: Obtain primitive $\boldsymbol{s^p} = \boldsymbol{w_p}^\top \bar{\boldsymbol{S}}$.
13: /* Contrastive Primitive Loss */
14: Obtain normalized similarity $d_{i,j}^y$ and softmax primitive similarity $d_{i,j}^s$ for image sample $x_i, x_j$.
15: Obtain primitive loss $L_p$.

---

## G  VISUALIZATION

This section investigates how the CompSLOT framework enhances continual learning performance by first demonstrating through slot attention mask visualizations across various benchmarks that CompSLOT successfully identifies important concepts (primitives) in an unsupervised manner, and then by presenting similarity matrix visualizations of ground truth concepts/primitives/features/logits for specific algorithms to illustrate the regularization effects that improve model compositional generalization and stability during continual learning. We attribute this robustness to "concept rehearsal": although class labels change, many visual concepts are shared and recur across tasks, helping stabilize the primitive selection weights. This is also discussed in Lai et al. (2024).

**Concept learning**  We evaluate CompSLOT on CGQA, COBJ, ImageNet-R, and CIFAR100 benchmarks by randomly selecting three representative images from each class. The extracted slot masks are visualized in Figures 5, 6, 7, and 8, respectively.

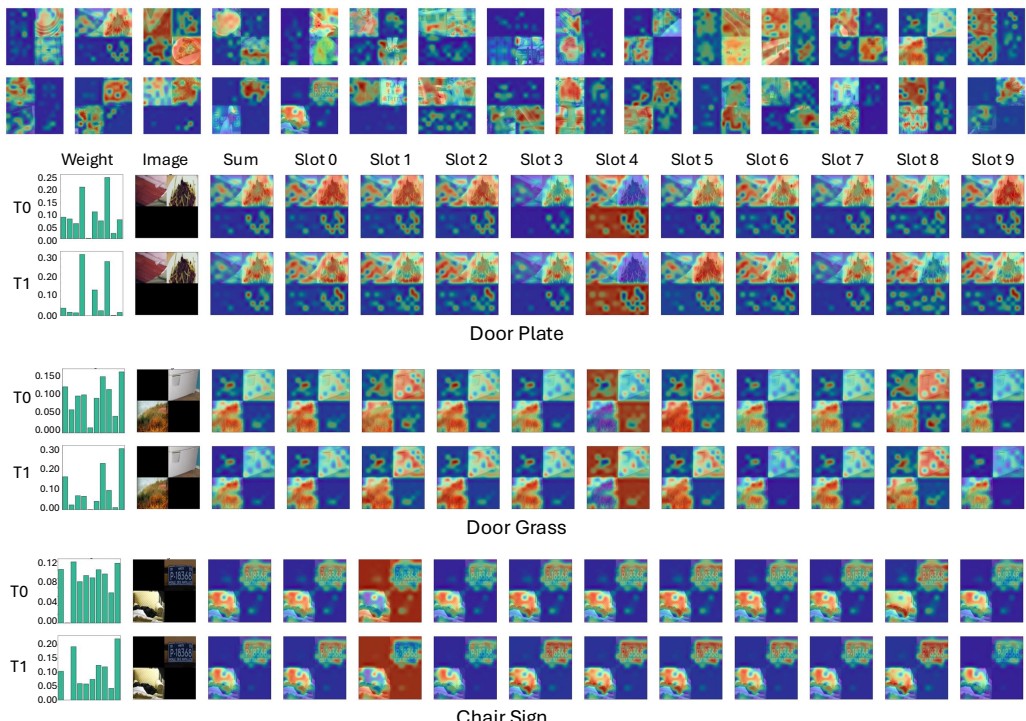

Figure 5: Visualization of learned slots on 30 randomly sampled images (3 images for each class in the first task of the 10-10 tasks) on CGQA. **Top row**: Primitives (weighted-sum of slot masks weighted by $w_p$) for 30 images. **Bottom 3 rows**: Three examples of images from classes (Door Plate), (Door Grass), and (Chair Sign) after being trained on the first task (T0) and on the second task (T1). **From left to right**: $w_p$, origin image, primitive (weighted-sum of slot masks), and 10 slot masks, respectively. **Takeaway**: CompSLOT successfully extracts primitives without any concept label.

On CGQA, the weighted slot masks (using weights $w_p$) effectively localize class-relevant concepts in each image. For instance, in the *Door Plate* class, slot 7 consistently captures the *Plate* concept while slot 8 focuses on the *Door*, demonstrating precise concept disentanglement. Notably, the learned primitives maintain visual consistency across tasks, that the primitive representation after task T0 closely resembles that after T1, confirming the stability of CompSLOT. This phenomenon was similarly observed in Figure 2.

The more challenging COBJ benchmark presents similar results. For an image in the *Other-shoes Person* class, slot 5 accurately identifies the *Other-shoes* concept while slot 7 correctly localizes the *Person*, even in this complex compositional setting.

When examining ImageNet-R and CIFAR100 with $K = 5$ slots, we observe that the primary concept corresponding to each class label is reliably identified, and the representations maintain discriminative power while preserving semantic consistency. However, the concept sharing is visually rare between classes, as demonstrated by the distinct slot activation patterns for different classes.

**Primitive-logit alignment** We conduct in-depth visualization analysis to understand the performance improvement of CompSLOT on COBJ, using ADAM + adapter as a representative example. We visualize 30 images for T0 and 60 images for T1 (10 old classes and 10 new classes). Figure 9 presents the cosine similarity matrix visualizations including: (a) Ground truth multi-hot concepts; (b) Extracted primitives; (c) Feature representations; (d) Final logits. The red boxes highlight two pairs of classes with concept sharing: (*Other-shoes Person*) and (*Other-shoes Person Sneaker*), as well as (*Person Sneaker*) and (*Other-shoes Person Sneaker*). CompSLOT successfully captures these shared concepts in the primitive representations (Figure 9b) and effectively distills them into the final

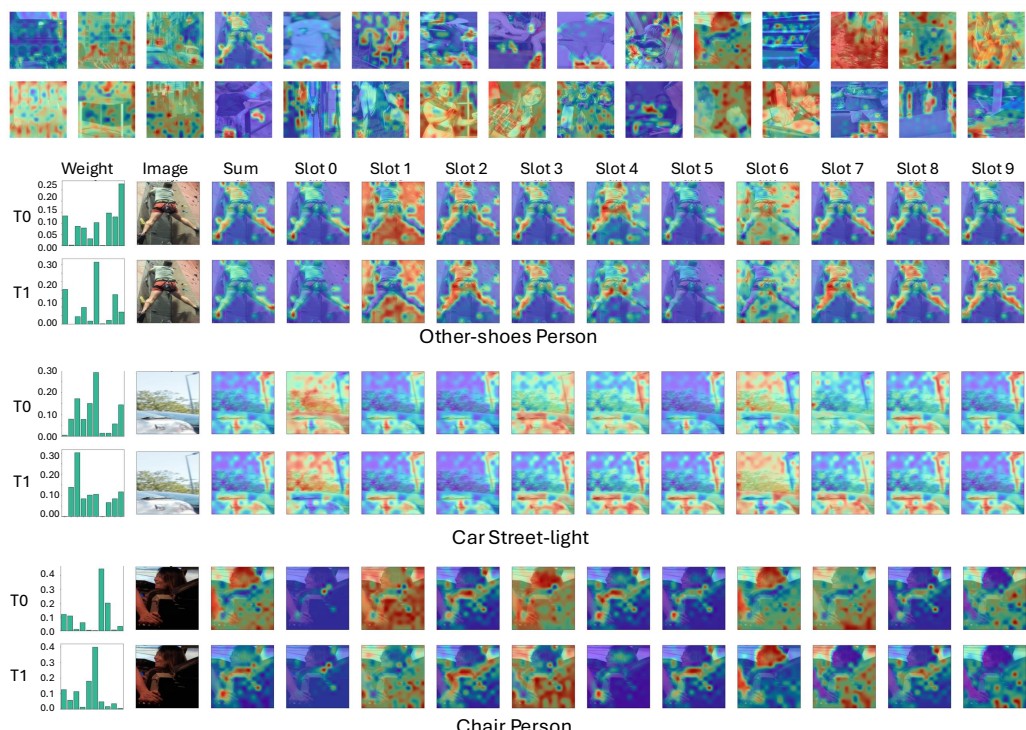

Figure 6: Visualization of learned slots on 30 randomly sampled images (3 images for each class in the first task of the 10-10 tasks) on COBJ. **Top row**: Primitives (weighted-sum of slot masks weighted by $w_p$) for 30 images. **Bottom 3 rows**: Three examples of images from classes (Door Plate), (Door Grass), and (Chair Sign) after being trained on the first task (T0) and on the second task (T1). **From left to right**: $w_p$, origin image, primitive (weighted-sum of slot masks), and 10 slot masks, respectively. **Takeaway**: CompSLOT successfully extracts primitives without any concept label.

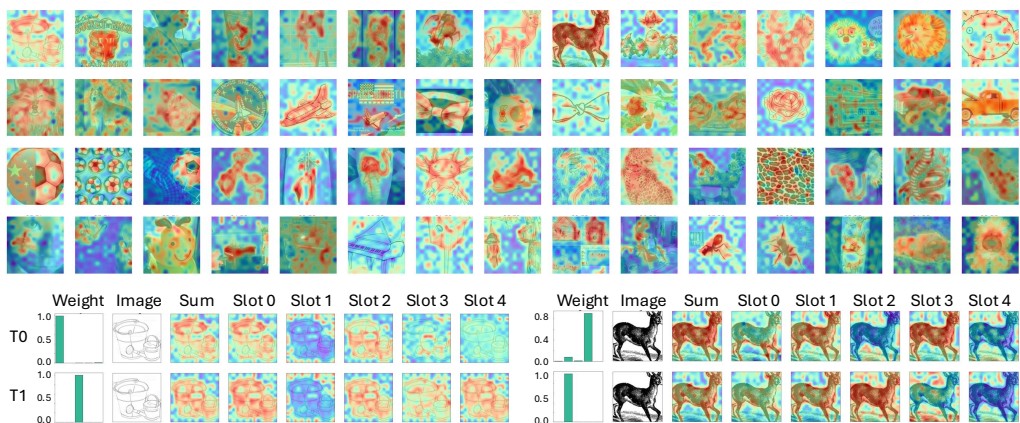

Figure 7: Visualization of learned slots on 60 randomly sampled images (3 images for each class in the first task of the 20-20 tasks) on ImageNet-R. **Top row**: Primitives (weighted-sum of slot masks weighted by $w_p$) for 60 images. **Bottom row**: Two examples of images after being trained on the first task (T0) and on the second task (T1). **From left to right**: $w_p$, origin image, primitive (weighted-sum of slot masks), and 5 slot masks, respectively. **Takeaway**: CompSLOT successfully extracts primitives without any concept label, and the concept sharing is rare between classes.

logits (Figure 9d). Notably, this alignment process also induces regularization at the feature level, as evidenced by the more coherent feature representations shown in Figure 9c.

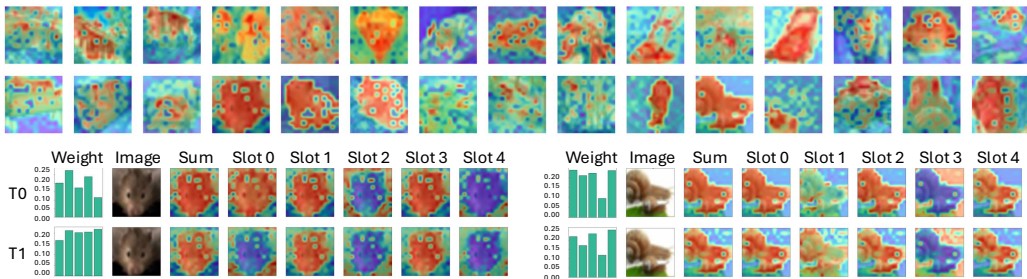

Figure 8: Visualization of learned slots on 30 randomly sampled images (3 images for each class in the first task of the 10-10 tasks) on CIFAR100. **Top row**: Primitives (weighted-sum of slot masks weighted by $w_p$) for 30 images. **Bottom row**: Two examples of images after being trained on the first task (T0) and on the second task (T1). **From left to right**: $w_p$, origin image, primitive (weighted-sum of slot masks), and 5 slot masks, respectively. **Takeaway**: CompSLOT successfully extracts primitives without any concept label, and the concept sharing is rare between classes.

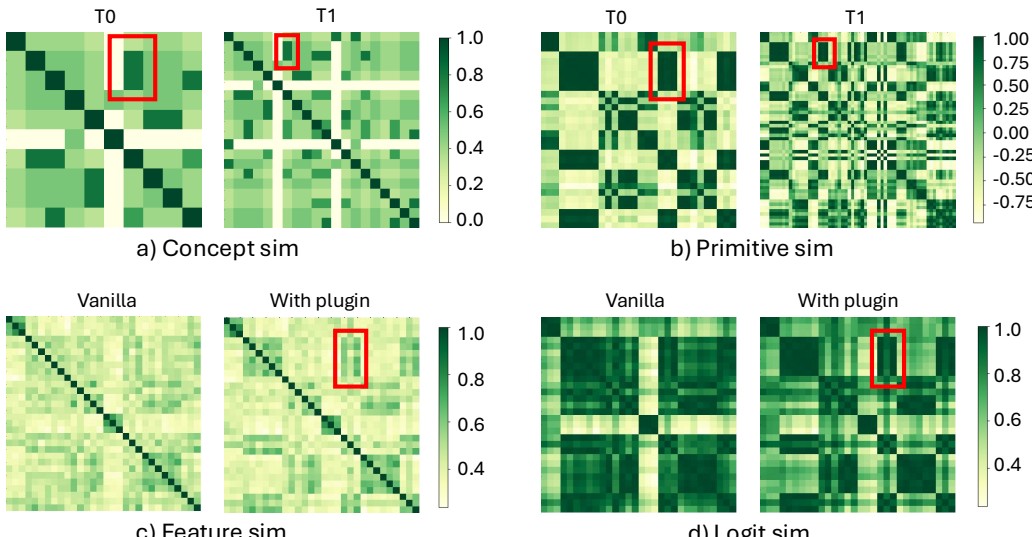

Figure 9: Visualization of a) concept; b) primitive; c) feature; d) logit cosine similarity matrices on sampled images (three images for each class in the first task T0 and second task T1 of the 10-10 tasks) on COBJ. a) **Left**: Multi-hot concept cosine similarity matrix of 30 images for T0; **right**: Multi-hot concept cosine similarity of 60 images (from the first-2 tasks T0 and T1). b) The primitive cosine similarity of the corresponding images. We use the learned pair-wise primitive similarity to mimic the statistics of the pair-wise concept similarity and regularize logits. c) **Left**: The learned feature cosine similarity matrix of 30 images in T0 for ADAM + adapter; **right**: The learned feature cosine similarity matrix of 30 images in T0 for ADAM + adapter †. d) The logit cosine similarity of the corresponding images as in c). **Takeaway**: The learned primitive successfully mimics concept statistics without concept supervision, and our $L_a$ successfully distills pair-wise primitive similarity into logits and affects the feature representations (as demonstrated with the regions marked with red box), while ADAM + adapter does not capture this concept sharing statistic.

We further validate CompSLOT on ImageNet-R, a standard CL benchmark without ground truth concept labels. Figure 10 shows the case performing CompSLOT on FOSTER. Our slot attention mechanism identifies shared concepts across six images (highlighted in red boxes), particularly revealing a consistent "*Fabric*" concept (Figure 10a). This automatic discovery of hidden relationships demonstrates CompSLOT's ability to generalize concept learning across different benchmarks.

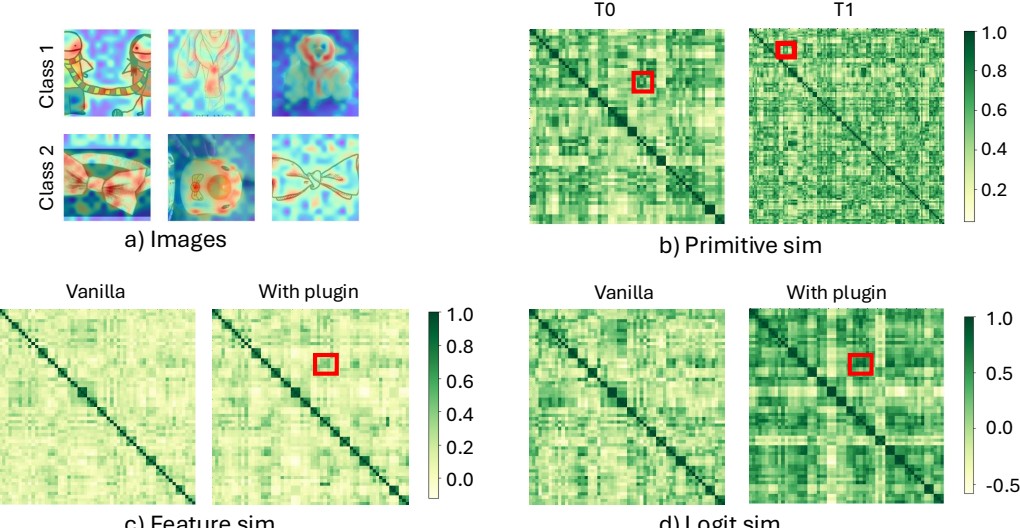

Figure 10: Visualization of a) images related to red box; b) primitive; c) feature; d) logit cosine similarity matrices on sampled images (three images for each class in the first task T0 and second task T1 of the 20-20 tasks) on ImageNet-R. a) Six images from two classes in T0 which corresponding to the red box. b) The primitive cosine similarity of the corresponding images. c) **Left**: The learned feature cosine similarity matrix of 60 images in T0 for FOSTER; **right**: The learned feature cosine similarity matrix of 60 images in T0 for FOSTER †. d) The logit cosine similarity of the corresponding images as in c). **Takeaway**: The learned primitives show that CompSLOT discovers hidden relationships based on concept as demonstrated with the regions marked with red box, while FOSTER does not capture this concept sharing statistic.

The consistent performance improvements reported in Sections 5 and H.3 validate that CompSLOT effectively captures meaningful semantic relationships, leading to better generalization and compositional learning capabilities.

# H  ADDITIONAL RESULTS

## H.1  DETAIL CFST RESULTS

The statistical analysis of each compositional test suite for the 10-10 tasks CGQA benchmark is presented in Table 5. All accuracy metrics are reported with their corresponding ±95% confidence intervals to quantify statistical significance. The key metrics include:

- Hn: Harmonic mean of compositional testing metrics (systematicity **sys**, productivity **pro**, substitutivity **sub**);
- Hr: Harmonic mean of reference testing metrics (Non-novel **non**, Not compositional **noc**);
- Ha: Harmonic mean across all test types;
- R=Hn/Hr: Ratio measuring compositional generalization improvement.

The results consistently demonstrate superior performance in both R and Hn (except for DER †), confirming CompSLOT's ability to enhance compositional generalization, particularly for systematicity and productivity properties. This aligns with our hypothesis that the slot plugin improves compositional reasoning capabilities. However, as previously reported in Liao et al. (2024) for ViT-based architectures, we observe no significant improvement in substitutivity, suggesting inherent limitations of ViT feature extractors in dealing with attribute shifting (e.g., color).

## H.2  INFLUENCES OF HYPERPARAMETERS

In this section, we investigate the effect of the introduced hyperparameters in the slot module w.r.t. the slot extraction performance and in the primitive-logit alignment loss. Without loss of generality,

Table 5: Detail CFST results. We report the average with ± 95% confidence interval.

| Methods | sys | pro | sub | Hn |
|---|---|---|---|---|
| CPrompt | 73.933±1.552 | 75.367±1.014 | **85.967±0.858** | 78.063±0.817 |
| CPrompt † | **75.133±1.835** | **78.133±0.971** | 84.600±0.514 | **79.091±1.086** |
| ADAM + adapter | 63.400±0.244 | 68.667±0.838 | 74.833±0.107 | 68.649±0.259 |
| ADAM + adapter † | **68.533±0.962** | **75.033±0.533** | **80.400±0.092** | **74.335±0.572** |
| RanPAC | 74.867±0.912 | 78.567±0.509 | **83.667±1.536** | 78.868±0.918 |
| RanPAC † | **75.833±1.764** | **80.600±0.800** | 83.433±1.783 | **79.815±0.829** |
| EASE | 74.900±0.423 | 80.567±0.629 | 84.233±0.282 | 79.713±0.449 |
| EASE † | **78.267±0.509** | **84.633±0.509** | **86.200±0.480** | **82.887±0.320** |
| CoFiMA | 83.100±1.135 | 86.767±0.267 | 90.600±0.606 | 86.711±0.483 |
| CoFiMA † | **84.467±0.324** | **88.967±0.373** | **91.767±0.141** | **88.297±0.278** |
| FOSTER | 86.900±0.514 | 91.400±0.489 | **91.233±0.971** | 89.791±0.086 |
| FOSTER † | **87.600±0.606** | **91.733±0.979** | 90.500±0.733 | **89.910±0.710** |
| DER | **87.700±0.160** | **91.733±0.838** | **91.033±0.828** | **90.119±0.510** |
| DER † | 86.567±0.509 | 90.300±0.666 | 90.200±0.320 | 88.986±0.129 |
| MEMO | 78.233±2.189 | 82.500±1.201 | 87.033±0.541 | 82.425±1.282 |
| MEMO † | **79.733±1.248** | **85.133±1.816** | **87.533±1.432** | **84.003±1.451** |

| Methods | non | noc | Hr | R |
|---|---|---|---|---|
| CPrompt | 76.400±0.973 | 86.033±0.437 | 80.926±0.360 | 0.964 |
| CPrompt † | **77.167±0.681** | **86.533±0.601** | **81.580±0.407** | **0.969** |
| ADAM + adapter | 66.167±0.930 | 82.867±0.615 | 73.580±0.809 | 0.932 |
| ADAM + adapter † | **71.267±0.417** | **84.967±0.192** | **77.516±0.323** | **0.958** |
| RanPAC | 75.267±1.063 | **80.033±0.833** | 77.574±0.813 | 1.016 |
| RanPAC † | **75.600±0.606** | 79.133±1.593 | 77.314±0.440 | **1.032** |
| EASE | 76.400±0.666 | 83.967±0.141 | 80.004±0.420 | 0.996 |
| EASE † | **79.900±0.185** | **85.867±0.541** | **82.775±0.255** | **1.001** |
| CoFiMA | 83.367±0.594 | 88.233±0.509 | 85.729±0.353 | 1.011 |
| CoFiMA † | **85.600±0.733** | **89.233±0.385** | **87.378±0.544** | **1.017** |
| FOSTER | **89.833±0.141** | **76.433±0.557** | **82.592±0.285** | 1.087 |
| FOSTER † | 89.700±1.543 | 68.767±2.199 | 77.847±1.992 | **1.154** |
| DER | **89.967±0.373** | **77.800±1.619** | **83.433±0.837** | 1.080 |
| DER † | 88.600±0.370 | 74.867±1.536 | 81.151±0.976 | **1.096** |
| MEMO | 80.533±1.802 | **79.600±0.489** | **80.053±0.790** | 1.029 |
| MEMO † | **82.433±2.214** | 77.700±2.080 | 79.985±1.847 | **1.050** |

Table 6: Varing $\beta$ results on CPrompt 10-10 CGQA (the first three tasks).

| $\beta$ | 0 | 0.1 | 0.5 | 1 | 2 | 5 | 10 | 50 |
|---|---|---|---|---|---|---|---|---|
| AA (%) ↑ | 68.33 | 68.43 | 69.67 | 70.17 | **70.87** | 70.40 | 70.13 | 69.00 |

(a) Primitive Loss Coefficient  (b) Primitive Loss Temperature  (c) Slot Selection Temperature

Figure 11: Radars of different hyperparameters in slot representation learning.

we report the model performance after training on the second task of 10-10 tasks CGQA in this section.

**Metrics** We learn slot representation $S$, attention mask $A$, and primitive representation $s^t$ as intermediate products of the forwarding process. Thus, it is necessary to design quantitative metrics to represent the performance of the learned slot as follows:

- **Primitive-label matching score: f1** $= -\text{MAE}(d^s, d^y)$, where $d^s$ and $d^y$ are described in Equation 3.
- **Primitive-concept matching score: f2** $= -\text{MAE}(d^s, d^c)$, where $d^c$ is similar with $d^y$ but the one-hot label is replaced with the multi-hot concept label. Note that the concept label is only used to analyze the performance of the learned slots and is never seen during training.
- **Task-wise matched attention mask mIoU: f3** $= \text{Mean}_t\{\text{IoU}(\mathcal{A}^\circ_{t-1}, \mathcal{A}^\circ_t)\}$, where $\text{IoU}(\cdot, \cdot)$ is the intersection over union metric and $\mathcal{A}^\circ_{t-1}, \mathcal{A}^\circ_t$ are matched attention sets (by Hungarian algorithm) extracted from the same image by the learners trained after task $t-1$ and $t$, respectively.
- **Task-wise weighted attention mask mIoU: f4** $= \text{Mean}_t\{\text{IoU}(w_{p,t-1}\top A_{t-1}, w_{p,t}^\top A_t)\}$.
- **Task-wise matched slot matching score: f5** $= -\text{MAE}(\mathcal{S}^\circ_{t-1}, \mathcal{S}^\circ_t))$.
- **Task-wise primitive matching score: f6** $= -\text{Mean}_x\{\text{MAE}(s^{t-1}_x, s^t_x)\}$.

For clarity, the matching scores are normalized to $[0, 1]$ to align with the range of mIoU. A large value of any metric above indicates a better performance according to the corresponding assessment.

**Slot representation learning** First, fixing $\tau_p = 100, \tau_t = 100$, we vary the coefficient $\alpha$ as shown in Figure 11a. While smaller $\alpha$ values (e.g., 0.1) achieve marginally better f6 scores (indicating greater primitive stability across tasks), they significantly degrade other critical metrics, particularly f1 and f2. This trade-off suggests that excessively stable primitives may fail to adequately capture diverse label semantics necessary for effective primitive-logit alignment.

Next, we examine the temperature parameter $\tau_p$ by fixing $\alpha = 10, \tau_t = 100$ (Figure 11b). The radar chart demonstrates that $\tau_p = 100$ provides optimal balance across all metrics, confirming our hypothesis that moderate temperature settings enable better concept generalization while preventing over-regularization.

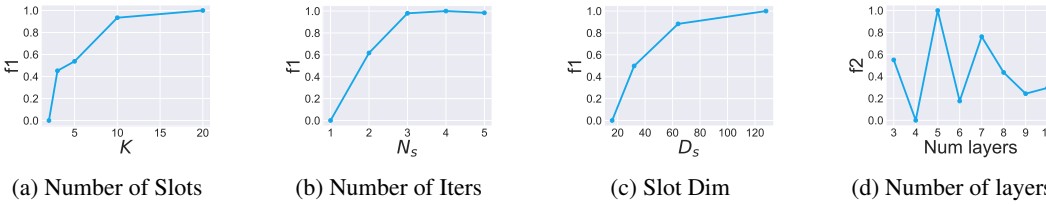

Figure 12: Line charts of different hyperparameters in slot attention architecture.

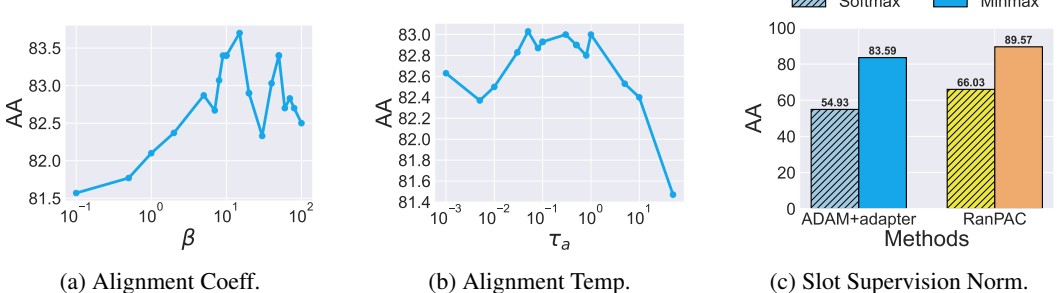

Figure 13: Line charts of different hyperparameters in primitive-logit knowledge distillation.

Finally, we analyze the task temperature $\tau_t$ with fixed $\alpha = 10, \tau_p = 100$ (Figure 11c). While no single $\tau_t$ value dominates across all metrics, we observe that $\tau_t = 100$ achieves the highest f1 score. Section G provides $w_p$ visualizations showing that larger $\tau_t$ values produce sharper slot selection distributions for primitive construction, which benefits concept representation but may reduce flexibility in extreme cases.

**Slot attention architecture**    Figure 12a examines the impact of increasing the number of slots ($K$). While higher $K$ values initially improve slot performance by enabling representation of more concepts, we observe diminishing returns beyond $K = 10$. This saturation occurs due to two factors: (1) the limited number of visually discriminable concepts per image, and (2) the finite capacity of the slot attention mechanism. Redundant slots tend to converge to similar representations, creating a performance plateau. Our slot mask visualizations in section G confirm this phenomenon, showing that excessive slots merely replicate existing patterns rather than capturing novel information.

Figure 12b investigates the effect of refinement iterations ($N_s$) in the slot attention module. While increasing $N_s$ enhances slot discriminability by promoting greater inter-slot differences, we find that three iterations ($N_s = 3$) achieve optimal performance. Further increases do not meaningfully improve results, suggesting that three iterations strike an effective balance between refinement and computational efficiency.

Figure 12c explores the relationship between slot dimensionality (capability) and performance. We observe that larger slot dimensions consistently improve f1 scores, indicating better concept representation. However, this comes at the cost of increased computational overhead, necessitating careful trade-off considerations for practical applications.

Finally, Figure 12d examines the impact of decoder architecture by varying MLP layer depth. Contrary to expectations, deeper decoders fail to improve extraction performance, suggesting that the current decoder architecture has sufficient capacity for the task.

**Primitive-Logit knowledge distillation**    We apply our learned slot attention mechanism to compute concept-based sample-wise similarities on RanPAC, systematically evaluating key hyperparameters in our primitive-logit knowledge distillation framework.

Figure 13a demonstrates that increasing the coefficient $\beta$ for $L_a$ consistently improves CL accuracy (AA). This indicates that stronger self-supervision from concept-based similarity effectively enhances the model's ability to preserve task-specific knowledge while adapting to new tasks.

Table 7: Main result on 10-10 tasks COBJ. We report the average accuracy after training the last task (AA), the cumulative average accuracy for each task (CA), and the final forgetting (FF). For CFST, we report the Harmonic mean of compositional testing (Hn) and the ratio of Hn and reference testing (R). Methods with CompSLOT are denoted with a postfix "†". Methods rehearse old samples are denoted with a postfix "*". We report results over 3 trials with (mean $\pm$ 95% confidence interval).

| Methods | Continual | | | CFST | |
| | AA (%) ↑ | CA (%) ↑ | FF (%) ↓ | Hn (%) ↑ | R↑ |
|---|---|---|---|---|---|
| CPrompt | 42.015±0.118 | 51.172±9.718 | 22.575±6.479 | 58.961±0.409 | 0.878 |
| CPrompt † | **45.520±0.421** | **52.565±0.931** | **19.575±1.029** | **59.880±2.032** | **0.880** |
| ADAM + adapter | 45.750±0.346 | 52.800±6.121 | 12.175±1.836 | 57.793±1.388 | 0.914 |
| ADAM + adapter † | **50.150±0.249** | **57.767±5.461** | **11.050±1.802** | **61.581±1.399** | **0.938** |
| RanPAC | 59.285±2.377 | 66.203±4.186 | **7.450±0.624** | 60.909±3.240 | 0.882 |
| RanPAC † | **61.950±0.527** | **67.367±4.075** | 7.875±0.104 | **62.317±2.447** | **0.889** |
| CoFiMA | 57.330±0.139 | **64.252±5.763** | 17.375±0.035 | **66.998±2.112** | 0.890 |
| CoFiMA † | **57.435±0.101** | 63.462±0.599 | **16.650±0.207** | 66.232±2.497 | **0.898** |
| FOSTER* | 47.800±0.542 | 53.741±0.290 | **10.575±0.759** | 62.750±0.337 | 0.852 |
| FOSTER* † | **50.980±0.225** | **59.735±0.556** | 14.525±0.240 | **63.695±0.312** | **0.908** |
| DER* | 55.815±0.714 | 64.905±3.342 | **23.650±2.425** | 68.558±0.189 | 0.844 |
| DER* † | **56.813±1.808** | **66.393±3.904** | 25.800±4.534 | **68.586±0.441** | **0.872** |

Figure 13b highlights the critical importance of properly tuning the temperature parameter $\tau_a$. We observe a performance plateau when $\tau_a$ is within an optimal range (approximately [0.1, 1.0]). Values beyond this range exhibit clear trade-offs. This is because (1) Large $\tau_a (> 1.0)$ causes excessive emphasis on sample-wise differences, undermining concept sharing; (2) Small $\tau_a (< 0.1)$ produces overly smooth logit similarities, degrading classification performance.

Regarding normalization strategies on primitives (Equation 5 min-max *vs* Equation 3 softmax), Figure 13c shows that min-max normalization outperforms softmax normalization. This advantage stems from min-max normalization's ability to provide sharper supervision through its linear scaling properties, and maintain better sensitivity to subtle concept differences between samples.

### H.3 RESULTS ON OTHER BENCHMARKS

**COBJ** The results in Table 7 clearly demonstrate that incorporating CompSLOT into CL methods with FMs leads to significant performance improvements across various metrics. Specifically, CompSLOT enhances compositional generalization ability, as evidenced by higher Hn and improved R (most significant gain of Hn for ADAM + adapter from 57.793 to 61.581), which in turn drives better overall CL performance (AA for ADAM + adapter improves from 45.75 to 50.15). CompSLOT's ability to strengthen compositional generalization appears to be the key factor behind these gains, enabling the model to better handle complex concepts and retain knowledge more effectively across tasks.

**ImageNet-R** It can be seen that CompSLOT can generally improve the performance of CL methods with FMs in Table 8. The improvement is likely due to the observation that the learned slot attention can discover hidden concept sharing between images, as evidenced by the visualization analysis in section G. Rehearsal methods (e.g., FOSTER* and MEMO*) achieve better performance in terms of AA and CA, comparing with rehearsal-free methods. This is because rehearsal methods can access old samples, thus, CompSLOT's primitive-logit alignment loss can provide more pairwise contrastive self-supervision on concept sharing, which enhances the model's compositional generalization performance.

Table 8: Main result on 20-20 tasks ImageNet-R. We report the average accuracy after training the last task (AA), the cumulative average accuracy for each task (CA), and the final forgetting (FF). Methods with CompSLOT are denoted with a postfix "†". Methods rehearse old samples are denoted with a postfix "*". The data for methods with citations is reported from the original paper. We report results over 3 trials with (mean $\pm$ 95% confidence interval).

| Methods | AA (%) ↑ | CA (%) ↑ | FF (%) ↓ |
|---|---|---|---|
| CPrompt (Gao et al., 2024) | 74.790±0.280 | **81.460±0.930** | 7.340±0.650 |
| CPrompt † | **75.225±0.270** | 79.964±1.078 | **6.989±1.126** |
| RanPAC | 78.375±0.062 | 82.519±0.839 | **4.856±0.367** |
| RanPAC † | **78.550±0.346** | **82.900±0.747** | 5.294±0.039 |
| CoFiMA | 80.025±0.146 | 83.927±1.421 | 7.614±0.142 |
| CoFiMA † | **80.250±0.016** | **84.118±1.017** | **7.022±0.005** |
| FOSTER* | 76.001±0.243 | 80.974±1.083 | **2.259±0.526** |
| FOSTER* † | **78.950±0.201** | **82.392±1.308** | 2.608±0.720 |
| MEMO* | 64.200±1.109 | 72.118±0.074 | **4.967±0.074** |
| MEMO* † | **65.200±0.249** | **72.995±1.251** | 5.344±0.256 |

Table 9: Varying backbone on 10-10 tasks CGQA. We report the average accuracy after training the last task (AA), the cumulative average accuracy for each task (CA), and the final forgetting (FF). The candidate CL algorithm is RanPAC. Methods with CompSLOT are denoted with a postfix "†"

| Backbone | AA (%) ↑ | CA (%) ↑ | FF (%) ↓ | Hn ↑ |
|---|---|---|---|---|
| ViT-B16 | 65.81 | 75.50 | 10.51 | 78.86 |
| ViT-B16 † | 66.75 | 76.58 | 10.21 | 79.81 |
| ViT-B16-DINO † | 66.58 | 76.62 | 10.24 | 80.39 |
| ViT-B16-SAM † | 67.30 | **77.76** | **9.67** | **81.22** |
| ViT-L16 † | **67.11** | 77.54 | 9.85 | 80.82 |

Table 10: AA results on 10-10 tasks CUB200 and CIFAR100.

| Datasets | SACK | CLG-CBM | CompSLOT |
|---|---|---|---|
| CUB200 | 71.78 | 85.40 | **88.38** |
| CIFAR100 | 87.26 | 84.49 | **89.57** |

## H.4 RESULTS ON OTHER BACKBONES

This section is to answer: **Do better vision foundation models contribute to better concept learning and continual learning performance?** We investigate the effect of model scaling via increasing the size and depth of the ViT architecture (e.g., ViT-L16 vs ViT-B16), and the effect of pretraining strategy via leveraging greater pretraining objectives, such as DINO (Oquab et al., 2024) (e.g., ViT-B16-DINO) and SAM (Kirillov et al., 2023) (e.g., ViT-B16-SAM), which have been shown to enhance semantic understanding, especially on segmentation and concept-rich tasks. We conduct experiments along the two key dimensions above and report the results in Table 9. The results show that ViT-L16 with larger model sizes demonstrates stronger representation modeling capabilities compared to ViT-B16, thus further boosting the significance of our CompSLOT. ViT-B16-DINO and ViT-B16-SAM with greater pre-training objectives exhibit better compositionality in decomposing concepts and continual learning performance, as reflected by higher Hn values.

Table 11: Additional ablation results on CGQA.

| Methods | $L_p$ | $L_a$ | AA (%) ↑ | FF (%) ↓ |
|---|---|---|---|---|
| SimpleCIL | ✗ | ✗ | **36.16** | **13.9** |
| SimpleCIL | ✓ | ✗ | 24.71 | 22.93 |
| RanPAC | ✗ | ✗ | **65.81** | **10.51** |
| RanPAC | ✓ | ✗ | 41.59 | 11.87 |

Table 12: Results on Finetuning on 10-10 CGQA. We report the average with ± 95% confidence interval.

| Methods | AA (%) ↑ | CA (%) ↑ | FF (%) ↓ |
|---|---|---|---|
| Finetuning | $29.91 \pm 0.84$ | $49.04 \pm 0.39$ | $58.36 \pm 1.02$ |
| Finetuning † | $\mathbf{33.48 \pm 0.04}$ | $\mathbf{52.43 \pm 0.65}$ | $\mathbf{50.01 \pm 2.13}$ |

## H.5 COMPARING WITH OTHER CONCEPT LEARNING METHODS

This section compares CompSLOT with other concept learning methods, i.e., CLG-CBM (Yu et al., 2025) and SACK (Kundargi et al., 2025). We conduct experiments on 10-10 tasks CUB200 and CIFAR100 to show the superiority of CompSLOT with RanPAC. To achieve fair comparison, we replace the CompSLOT's backbone to CLIP ViT-B/16. The results are shown in Table 10 with the top performance mentioned in their original papers. CompSLOT shows the best AA on both benchmarks, because of benefiting from slot attention to extract concept information and the plug-and-play property that can be applied to alternative CL backbones. Most importantly, CompSLOT fully utilizes the capability of the CL backbone and does not need extra interpretable tools, like ChatGPT.

## H.6 ADDITIONAL ABLATION STUDIES

To clearly substantiate the contribution of slot attention in combination with primitive selection, we conduct an ablation study where we remove knowledge distillation and instead directly use the learned primitive representations with a cosine similarity classifier for continual tasks, as in SimpleCIL (Zhou et al., 2025). We also integrate this strategy into RanPAC and the results are shown in Table 11. This naive approach suffers from severe forgetting, confirming that primitive representations are insufficient for long-term retention when learning new tasks. In contrast, our alignment loss distills pair-wise relational information (i.e., a compact, low-dimensional encoding of concept combinations) rather than high-dimensional raw representations. This enables methods equipped with CompSLOT to maintain stable performance while accumulating higher accuracies over time, demonstrating the efficacy of CompSLOT in mitigating catastrophic forgetting.

## H.7 CASE STUDIES: THE EFFECT OF COMPSLOT ON FINETUNING

In this section, we answer that question: **Does CompSLOT itself benefit continual learners without associating with other continual learning algorithms?** We perform CompSLOT on a naive continual learner **finetuning**, which uses a frozen feature extractor backbone (ViT-B/16) and a extendable classifier. The results are shown in Table 12. We observe that CompSLOT successfully achieves higher AA with smaller FF. This observation indicates that CompSLOT itself, as a plug-in, benefits the continual learner without the need to combine other mechanisms.

## H.8 ALGORITHM EFFICIENCY ANALYSIS

**Parameter overhead** We evaluate the parameter overhead introduced by our slot attention module. As this module requires a pretrained ViT as its semantic feature extractor, which is a standard component in all continual learning of foundation models frameworks, the additional trainable

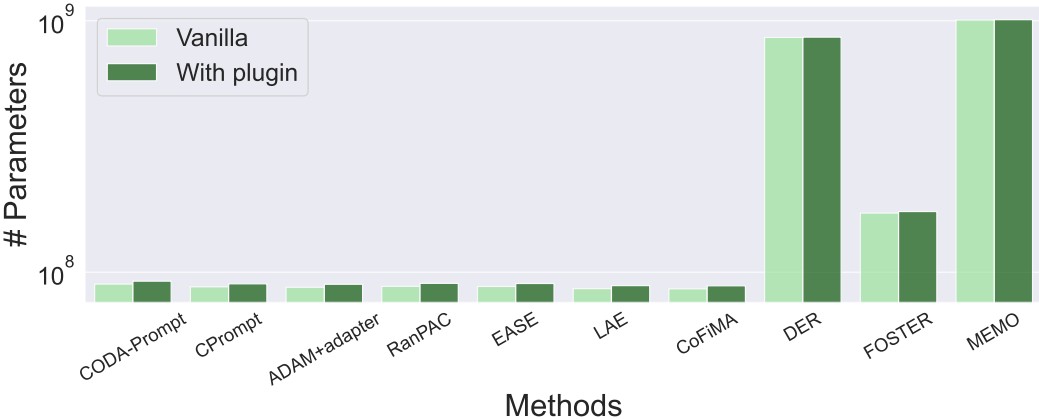

Figure 14: Visualization of the parameter numbers for methods with and without the slot module. Note that the data are collected according to the default implementation in the PILOT (Sun et al., 2025) platform and after the training of the last 10-way CGQA continual task. **Takeaway**: CompSLOT requires a ViT backbone that is already in any model-based continual learner with a foundation model, thus, it is light-weight and free to be applied.

Table 13: Computational overhead (h) on CGQA.

| Slot module | FOSTER | FOSTER † |
|:-----------:|:------:|:--------:|
| 5.5 | 9.1 | 10.1 |

parameters are negligible compared to the total model size, as illustrated in Figure 14. This makes our CompSLOT computationally efficient while delivering significant performance benefits.

**Computation overhead** In Table 13, we study the computational overhead introduced by the slot attention mechanism and primitive extraction. As an example, we choose FOSTER as a representative baseline, since it achieves nearly top performance among others. We compare three cases: 1) Continual training of just our slot module plugin, including both slot attention and primitive selection components, without applying it to other continual learning algorithms; 2) Full continual training of FOSTER; 3) Full continual learning of FOSTER with a pretrained slot module plugin (FOSTER †). We highlight that the slot module can be learned offline as a reusable component which only associated with the benchmark and is independent of algorithms. Once trained, it serves as a pretrained plugin that can be directly loaded for any continual learning algorithm with minimal additional overhead. It only requires adding alignment loss $L_a$ for logit regularization and spending an additional 10% of total training time for FOSTER from 9.1h to 10.1h. This design is particularly beneficial when running multiple continual learning algorithms on the same data distribution.

Importantly, we conduct an ablation study (Section 5), where we deliberately increase the parameter count of baseline CL methods to match our CompSLOT-enhanced models. The results demonstrate that the performance gains stem not from increased capacity, but from CompSLOT's improved compositional generalization capabilities. This confirms that CompSLOT provides genuine algorithmic advantages rather than simply benefiting from more parameters.

# I    USE OF LARGE LANGUAGE MODELS

In the process of preparing this paper, we employed LLMs to polish the writing of the paper. The assistance provided by LLMs was mainly focused on improving the clarity, coherence, and overall quality of the language used in the manuscript. We input sections of the paper into the LLM and requested it to suggest rephrasings, correct grammar and spelling errors, and enhance the readability of the text. It is important to note that LLMs did not play a significant role in the research ideation. The core ideas, research questions, experimental designs, and methodological choices were independently

conceived and developed by the human authors. The LLM was not involved in formulating the hypotheses, determining the research direction, or making decisions regarding the data collection and analysis methods.

