# OpenReview forum: "Plug-and-Play Compositionality for Boosting Continual Learning with Foundation Models"
_ICLR.cc/2026/Conference — ICLR 2026 Oral_

### Official Review · Reviewer_rGQP · 2025-10-26

**Soundness:** 2
**Presentation:** 3
**Contribution:** 2
**Rating:** 4
**Confidence:** 3

**Summary:**

This paper proposes a new continual learning algorithm for foundation models called CompSLOT. CompSLOT utilizes an existing object-centric plug in for concept extraction, and incorporates concept learning through decomposition and selection mechanism. Experiment results show that CompSLOT boosts several continual learning methods' performance on standard metrics and benchmarks.

**Strengths:**

1. The proposed method bridges object-centric learning, concept learning and continual learning, which is a novel intersection.
2. Extensive experiments have been conducted to confirm that CompSLOT improves continual learning performance on standard evaluation metrics and benchmarks.
3. The proposed method can be easily combined with other continual learning methods.
4. The paper is well-written and easy to follow.

**Weaknesses:**

1. **The models for experiments are not specified.** The authors do not mention of a specific architecture in the main paper. Does all experiments in Table 1 conducted with the same model architecture? Do CLG-CBM and CompSLOT use the same pretrained model? Without these details, it is difficult to confirm the effectiveness and the versatility of the proposed method.
2. **Lack of concept analysis.** The authors claim that CompSLOT learns human-interpretable concepts for continual learning. However, no experiments are presented to analyze these concepts or evaluate interpretability of the learned representations.
3. **Hyperparameter sensitivity.** CompSLOT requires hyperparameter tuning ($\alpha, \beta, \tau_t, \tau_p, \tau_a$), which is data-dependent. The sensitivity to these hyperparameters is not discussed in the paper.

**Questions:**

1. Does Slot Attention suitable for any ViT, or other vision models?
2. I was wondering if CompSLOT benefits the naive continual learners: finetuning? Or does CompSLOT need to be combined with other continual learning algorithms? It will be an informative baseline.

---

> ### Author Response · Authors · 2025-11-19
> **To Reviewer rGQP (1/2)**
>
> We sincerely appreciate your constructive comments on this paper. We detail our response below point by point. Please kindly let us know if our response addresses the issues you raised in this paper.
> Please note that the line numbers mentioned below, as well as the section numbers, figure and table numbers, are all based on the original submission. Their relative positions may change in the revision.
> ## W1: The models for experiments are not specified
> > - **We sincerely appreciate the reviewer's insightful observation regarding architectural clarity.** To ensure full reproducibility and methodological transparency, we provide the following clarifications:
> >     - All experiments in **Table 1 & 2** utilize the **ImageNet-21k pretrained ViT-B/16** architecture from the timm Python package. This choice was explicitly documented in *Appendix E, Lines 1019-1020*.
> >     - Our method, CLG-CBM, and the other baselines  in **Table 3** all employ **CLIP ViT-B/16**.
> >     - Furthermore, we investigate the effect of different backbones (e.g., ViT-L/16) in *Appendix J, Table 9*.
> > - We appreciate the reviewer's identification of the missing details in the main paper. We assure that we will modify our new revision:
> >     - provide **these information in the experimental setting section**;
> >     - include **complete details in Appendix E**;
> >     - **highlight in the experimental setting section**: "for the details of experimental settings, please refer to Appendix E".
> ## W2: Lack of concept analysis
> > - **We agree the concept analysis is very important.** Thus, we made the following efforts:
> >     - We conducted experiments to **visualize the learned concepts** in *Appendix K*. This information was highlighted in the main paper at *Line 299* (right after introducing how we learn primitives) and in the experimental part at *Line 335*.
> >         - We also visualized one example in *Figure 1*.
> >     - Furthermore, to evaluate whether the learned concepts align with the ground truth, we established **numerical experiments** in *Appendix H* and designed **six metrics** to assess concept performance, including mask mIoU and slot representation MAE.
> >     - We also visualized and **compared ground truth concept similarity with primitive similarity** in *Appendix K, Figures 11 and 12*.
> > - Here we briefly list some observations for your interests:
> >     - CompoSLOT successfully identifies **important concepts (primitives)** in an **unsupervised** manner.
> >     - The learned **primitives** successfully mimic **concept statistics** without **concept supervision**.
> >     - The proposed *primitive-logit alignment loss* successfully distills **pair-wise primitive similarity** into **logits** and affects the **feature extraction**, which demonstrate that the models make decisions by additionally considering **low-dimensional concept combinations** instead of only relying on **high-dimensional features**.
> >     - We also attribute this robustness to *concept rehearsal*: although class labels change, many visual concepts are shared and recur across tasks, helping stabilize the primitive selection weights, as mentioned in *Lines 301-303*.
> > - **We appreciate that we did not provide a throughout discussion about the concept analysis in the main paper** due to page limit. We promise that, in our new revision, we will:
> >     - provide a paragraph specifically discussing the **learned concepts** in the *experimental part*.

---

> > ### Author Response · Authors · 2025-11-19
> > **To Reviewer rGQP (2/2)**
> >
> > ## W3: Hyperparameter sensitivity
> > > - **We agree that the analysis of hyperparameters is a basic part of our proposed method.** Thus, we made the following efforts:
> > >     - We established experiments in *Appendix H* to thoroughly study the effect of every hyperparameter.
> > >     - Due to page limit, we chose to highlight the conducted experiments in the *main paper experimental part Line 334*.
> > > - For your interests, we investigated hyperparameters introduced in three mechanisms: *concept learning, slot attention architecture, and primitive-logit knowledge distillation*. Specifically:
> > >     - The impact of **primitive loss coefficient $\alpha$** was shown in *Appendix Figure 4a* and discussed in *Lines 1219-1222*.
> > >     - The impact of **primitive-logit alignment loss coefficient $\beta$** was shown in *Appendix Figure 6a* and discussed in *Lines 1256-1258*.
> > >     - The impact of **slot selection temperature $\tau_t$** was shown in *Appendix figure 4c* and discussed in *Lines 1227-1231*.
> > >     - The impact of **primitive loss temperature $\tau_p$** was shown in *Appendix figure 4b* and discussed in *Lines 1223-1226*.
> > >     - The impact of **primitive-logit alignment loss temperature $\tau_a$** was shown in *Appendix figure 6b* and discussed in *Lines 1259-1263*.
> > > - **We appreciate that we did not provide a throughout discussion about the hyperparameters we introduced in the main content.** We promise that, in our new revision, we will:
> > >     - provide a paragraph specifically discussing the **hyperparameter sensitivity** in the *experiments* part.
> >
> >
> > ## Q1: Does Slot Attention suitable for any ViT, or other vision models?
> > > - The Slot Attention module takes **semantic patch features** as input source. Thus, it is suitable for backbones which provide **patch-wise hidden representations**.
> > > - We choose **ViT** foundation backbones because it provides informative patch-wise features and the community has a rich family of released checkpoints, which makes it possible for us to conveniently investigate the effects of **different backbones**.
> > > - According to [1], we would like to highligh the efforts we made to explore the effects of different backbones as follows:
> > >     - We investigated the *different architectures of vision models* of **ViT-B/16** and **ViT-L/16**;
> > >     - and the *pretraining strategies* of **DINO** and **SAM**. Note that for the fairness of comparison, we choose the same backbone structure, which is ViT-B/16.
> > > - For the details of the experiments, please refer to our discussion in *Appendix J* and the results were shown in *Table 9*. For your convenience, we also show them in *Table 1* below. The results shows that:
> > >     - ViT-L16 with **larger model sizes** demonstrates stronger representation modeling capabilities compared to ViT-B16, thus further **boosting the significance of our CompoSLOT**.
> > >     - DINO and SAM with **greater pre-training objectives** exhibit better compositionality in decomposing concepts and continual learning performance, as reflected by **higher Hn values**.
> >
> > Table 1: Investigating different architectures.
> > | Methods | AA% $\uparrow$| CA% $\uparrow$ | FF% $\downarrow$ | Hn $\uparrow$|
> > | --- |--- | --- | --- | --- |
> > | ViT-B16 | 65.81 | 75.50 | 10.51 | 78.86 |
> > | ViT-B16 $\dagger$ | 66.75 | 76.58 | 10.21 | 79.81 |
> > | ViT-L16 $\dagger$ | 67.11 | 77.54 | 9.85 | 80.82 |
> > | DINO $\dagger$ | 66.58 | 76.62 | 10.24 | 80.39 |
> > | SAM $\dagger$ | 67.3 | 77.76 | 9.67 | 81.22 |
> >
> > [1] Awais M et al. Foundation models defining a new era in vision: a survey and outlook. PAMI, 2025.
> >
> > ## Q2: Whether CompSLOT benefits the naive continual learner: finetuning?
> > > - Since our primitive-logit alignment loss regularizes the output **logits of CL classifiers**, it is appropriate to apply our CompSLOT on **finetuning learners**.
> > >     - As you suggested, we conduct experiments on finetuning on *10-10 CGQA* over two trials. The results are shown in *Table 2* below.
> > >     - We observe that our plug-in successfully achieves higher **average task accuracy** with smaller **average forgetting**.
> > >     - This observation indicates that your expectation is correct: CompSLOT itself, as a plug-in, benefits the continual learner **without the need to combine other mechanisms**.
> > > - We appreciate your constructive question that finetuning does not contain other **method-specific mechanisms**, and thus directly reflects the benefit of our plug-in. **We will update the results and discussion in our revision.**
> >
> > Table 2: Results on finetuning.
> > | method | AA% $\uparrow$| CA% $\uparrow$| FF% $\downarrow$| R $\uparrow$ |
> > | --- | --- | --- | --- | --- |
> > | FT | 29.91 $\pm$ 0.84 | 49.04 $\pm$ 0.39 | 58.36 $\pm$ 1.02 | 1.09 |
> > | FT $\dagger$ | 33.48 $\pm$ 0.04 | 52.43 $\pm$ 0.65 | 50.01 $\pm$ 2.13 | 1.14 |
> >
> > **We sincerely appreciate your thoughtful questions, which has helped us better articulate important aspects of our work.** Please let us know if you would like us to provide any additional comments!

---

> > > ### Author Response · Authors · 2025-11-25
> > > **Appreciation for the Review and Willingness to Clarify Further**
> > >
> > > Dear Reviewer rGQP,
> > >
> > > Thank you very much for taking the time to review our work. We greatly appreciate your valuable feedback and suggestions. If there are any remaining questions or if any points in our rebuttal were unclear, please feel free to let us know. We are happy to further clarify and actively participate in the discussion. We will also carefully address the shortcomings in our paper and make improvements in our revised manuscript.
> > >
> > > Best regards,
> > >
> > > The Authors

---

> > > > ### Comment · Reviewer_rGQP · 2025-11-26
> > > >
> > > > Thank you for your response. Most of my concerns have been addressed. I think the paper is more complete now. According to Appendix H, the proposed method requires careful hyperparameter tuning. Nonetheless, I believe the paper has novel contributions to the continual learning community. Therefore, I have increased my score.

---

> > > > > ### Author Response · Authors · 2025-11-26
> > > > >
> > > > > Dear Reviewer rGQP,
> > > > >
> > > > > Thank you for the positive assessment and for raising the score. We are glad the paper is more complete in your view.
> > > > >
> > > > > Best regards,
> > > > >
> > > > > The Authors

---

### Official Review · Reviewer_hVcV · 2025-10-30

**Soundness:** 4
**Presentation:** 2
**Contribution:** 3
**Rating:** 6
**Confidence:** 2

**Summary:**

The proposed CompSLOT (Compositional Slot plug-in) is a novel method designed to address the challenges of continual learning on compositional benchmarks. It enhances vision models by extracting disentangled, class-relevant concepts directly from images using Slot Attention mechanism. The core of CompSLOT lies in its robust concept learning phase, which uses the primitive selection and aggregation mechanism to identify essential class-relevant concepts. Experimental results consistently demonstrate that CompSLOT significantly boosts the performance of various state-of-the-art continual learners across challenging compositional benchmarks.

**Strengths:**

- CompSLOT can significantly mitigate catastrophic forgetting in continual learning.

- CompSLOT features a highly flexible and method-agnostic plug-and-play design.

- Extensive experiments on challenging compositional datasets robustly validate CompSLOT's superior performance in continual learning.

**Weaknesses:**

- It is unclear how a fair comparison was achieved, as the compared methods were not used in the benchmark paper. It is also not understood how these Class-Incremental Learning methods were applied in this setting.

- Essentially, CompSLOT uses an external model as a teacher to reduce catastrophic forgetting.

- It is unclear how this plugin works with baseline methods. For instance, the plugin generates a set of logits. However, the contribution of some methods, such as CPrompt, is to constrain the logits at different stages during incremental learning. Would the proposed method conflict with these baselines?

- The method is too complex to be reproduced easily.

**Questions:**

see above.

---

> ### Author Response · Authors · 2025-11-19
> **To Reviewer hVcV (1/3)**
>
> We sincerely appreciate your constructive comments on this paper. We detail our response below point by point. Please kindly let us know if our response addresses the issues you raised in this paper.
> Please note that the line numbers mentioned below, as well as the section numbers, figure and table numbers, are all based on the original submission. Their relative positions may change in the revision.
>
> ## W1&W2: How to achieve fair comparison for the compared methods, and CompSLOT uses an external model as a teacher
> > - **We appreciate the reviewer’s insightful comments regarding the fairness of the comparisons**. We fully agree that fair comparison is essential. Therefore, in our experiments, the proposed plug-in module **does not** introduce any additional **external knowledge**, such as **concept supervision** or **extra pretrained weights**.
> > - Moreover, we provide detailed implementation settings in the paper to ensure that all methods are compared under consistent conditions, including:
> >     - When comparing with and without CompSLOT, e.g., in *Table 1*, we use exactly **the same backbone** for both **continual learners and the slot attention**, which is the *ViT-B/16 backbone pretrained on ImageNet-21K* sourced from the *Python timm package*, as we pointed out in *Lines 95-97* and in *Lines 1019-1020*.
> >     - When training slot attention, we **do not** introduce additional **supervision**, such as **concept labels**, as we pointed out in *Lines 90-91*.
> >     - For the additional hyperparameters introduced in this work, we employ an **80-20 train-validation split** using **validation sets**, as we mentioned in *Lines 1017-1018*.
> >     - And other hyperparameters use the **default settings**, as suggested in the **PILOT platform [1]**, as we mentioned in *Line 344*.
> >     - For the additional compositional testing in CGQA and COBJ, we use **randomly generated 300 few-shot tasks** for each test suite, as suggested in the **CFST benchmarking paper [2]**, as we mentioned in *Line 369*.
> >     - To further ensure fairness and show that **performance gains are not from the increased model capacity**, we also compare with a case extending the number of parameters in an ablation study (*Lines 438-444*).
> > - These measures collectively ensure that the comparison among different methods is as fair and rigorous as possible. Furthermore, we promise we will give our new revision:
> >     - provide a **thorough discussion** about the fairness issues in *Appendix B*,
> >     - improve *Appendix E* to **include all the implementation details**,
> >     - **highlight in the main paper** that "To show the efforts we make to achieve a fair comparison, please refer to the discussion in Appendix B and implementation details in Appendix E".
>
> [1] Hai-Long Sun, et al. "Pilot: A pre-trained model-based continual learning toolbox". SCIENCE CHINA Information Sciences 2025.
>
> [2] Weiduo Liao, et al. Does continual learning meet compositionality? new benchmarks and an evaluation framework. NeurIPS 2024.

---

> ### Author Response · Authors · 2025-11-19
> **To Reviewer hVcV (2/3)**
>
> ## W3: How does this plugin work with baseline methods?
> > - We would like to humbly clarify that CompSLOT does not generate a set of logits. Instead, CompSLOT works as follows:
> >     - The **concept learning** part provides **pair-wise concept-level similarities** between samples in the batch in an unsupervised way, and we distill such pair-wise similarities into the current task's **logits** (outputs of the classifiers) with a **contrastive loss**, as mentioned in *Lines 113-115* and detailedly described in *section 4.2*.
> >     - As a result, the model intentionally performs classification by referring to the **class-relevant concepts** (i.e., **primitives**) within samples.
> >     - As mentioned in *Lines 50-53*, understanding classes by decomposing them into **low-dimensional concept combinations** (rather than **high-dimensional feature combinations**, as in traditional methods) enables rapid adaptation on new tasks and mitigates forgetting on old tasks.
> >         - This is evidenced in our main *Table 1* (also in [3,4]) and visually demonstrated in *Appendix K (Lines 1549-1565)*.
> >     - We also attribute this robustness to *concept rehearsal*: although class labels change, many visual concepts are shared and recur across tasks, helping stabilize the primitive selection weights, thus benefit primitive recognition, as mentioned in *Lines 301-303*.
> > - **We sincerely appreciate the reviewer raising this question**, as it highlights the need for clearer communication of our method's key contributions in the current submission. We will **enhance the contribution sections accordingly in our revision**.
>
> [3] Lu Yu, et al. "Language guided concept bottleneck models for interpretable continual learning". CVPR 2025
>
> [4] Rymarczyk, Dawid, et al. "Icicle: Interpretable class incremental continual learning". ICCV 2023
>
> ## W3: Would the proposed method conflict CPrompt's constraining the logits [5]?
> > - We would like to answer this question point-by-point:
> >     - The answer depends on the value of coefficient $\beta$.
> >     - Regarding the *smooth regularization* proposed in CPrompt, its purpose is to reduce incorrectly activated logits for **previous tasks**.
> >     - For our plug-in, although the primitive-logit alignment loss primarily manipulates logits for **the current task**, it may inevitably **affect the logit values of correct labels** and indirectly **trigger CPrompt's smooth regularization**.
> >     - Therefore, the optimization goal is to achieve a **common agreement among these constraints**.
> >     - We conduct quick experiments by varying coefficient $\beta$ and examine its effects on continual learning performance. The results are shown in Table 1 below.
> >     - We observe that **average task accuracy** increases as $\beta$ increases but decrease after a threshold (around 2).
> >     - This indicates that an **excessively large $\beta$** hinders the effectiveness of CPrompt's smooth regularization, leading to conflicts. **However, within an appropriate range, our CompSLOT works effectively with CPrompt's smooth regularization** (as demonstrated in the *main paper Table 1*).
> > - **We sincerely appreciate the reviewer for this insightful suggestion**, and we will incorporate these observations in **one paragraph in experimerntal part of our new revision** specifically discusses the influence of hyperparameters.
>
> Table 1: CPrompt $\dagger$ on 10-10 CGQA (firsh three tasks)
> | $\beta$ | 0 (no CompSLOT) | 0.1 | 0.5 | 1 | 2 | 5 | 10 | 50 |
> | --- | --- | --- | --- | --- | --- | --- | --- | --- |
> | AA % $\uparrow$ | 68.33 | 68.43 | 69.67 | 70.17 | 70.87 | 70.4 | 70.13 | 69.00 |
>
> [5] Gao, Zhanxin, et al. "Consistent prompting for rehearsal-free continual learning." CVPR 2024.

---

> > ### Author Response · Authors · 2025-11-19
> > **To Reviewer hVcV (3/3)**
> >
> > ## W4: The method is too complex to be reproduced easily
> > > - **We sincerely appreciate the reviewer for highlighting the reproduction concern**. We would like to emphasize the concrete measures we will undertake:
> > >     - As committed in our response to **W1**, we will incorporate **as many implementation details as possible** in *Appendix E*.
> > >     - Our project builds upon the **implementation of the PILOT platform [6]** (as mentioned in *Line 344*), which provides a suite of continual learning algorithms integrated with foundation models.
> > >     - While we are unable to share the **full source code** during the review period due to **confidentiality constraints**, we will **provide a concise code draft for the wrapper implementing our CompSLOT** based on PILOT's learner in *Algorithm 1* below, presented in Python.
> > >         - Our CompSLOT is implemented via a `LearnerWrapper`, designed to be lightweight and applicable to any provided continual learning algorithm that exposes a method `_get_loss()` returning the `batched_loss` and `batched_logits`. The **primitive-logit alignment loss** is computed within the method `_prim_logit_reg()`.
> > >     - We commit to releasing the **complete codebase** along with a **comprehensive README** upon acceptance of our work.
> >
> >
> > Algorithm 1
> > ```python
> > class LearnerWrapper:
> >     """Overwrite _get_loss method of the wrapped PILOT learner to apply additional primitive-logit alignment loss.
> >     """
> >     def __init__(self, learner_instance):
> >         self.learner = learner_instance
> >         self._get_loss_ori = self.learner._get_loss
> >         self.learner._get_loss = self._get_loss
> >         # Overwrite learner._get_loss, thus, it will be cal-ed in learner's _init_train function.
> >
> >         self.args = self.learner.args
> >         self.slot_args = self.load_slot_args()
> >         self.slot_learner = Learner(self.slot_args)
> >         # Load slot model to provide primitive-logit aligment loss.
> >         self.slot_dim = self.slot_args['slot_dim']
> >
> >     def __getattr__(self, name):
> >         """Delegate attribute access to the wrapped Learner instance.
> >         """
> >         return getattr(self.learner, name)
> >
> >     def _get_loss(self, inputs, targets, *args, **kwargs):
> >         """IMPORTANT!!!
> >         Override the _get_loss method of the wrapped Learner.
> >         This will be call-ed in learner's method.
> >         """
> >         loss, res = self._get_loss_ori(inputs, targets, *args, **kwargs)
> >         # Forward the original _get_loss method to obtain logits to apply primitive-logit alignment loss.
> >
> >         logits = res['logits']
> >         # Should be the logits used to calculate the loss
> >
> >         _, slot_collect = self.slot_learner._network(inputs, pen=True)
> >         # slot_collect contains: 'slots', 'attns', ...
> >
> >         primitive = slot_collect['primitive']
> >         sl_loss = self._prim_logit_reg(primitive, logits)
> >         # IMPORTANT!!! calculate primitive-logit alignment loss
> >
> >         loss = loss + beta * sl_loss
> >         # beta is coefficient in the paper.
> >         res['prim_logit_align_loss'] = sl_loss
> >
> >         return loss, res
> >
> >     def incremental_train(self, data_manager):
> >         # load slot model before call learner.incremental_train method
> >
> >         task_id = self._cur_task + 1
> >         self.data_manager = data_manager
> >
> >         self.load_slot_checkpoint(task_id)
> >         # Load pre-learned slot.
> >
> >         self.learner.incremental_train(data_manager)
> >         # Apply continual training on the current task.
> >
> > ```
> >
> >
> >
> > [6] Hai-Long Sun, et al. "Pilot: A pre-trained model-based continual learning toolbox". SCIENCE CHINA Information Sciences 2025.
> >
> > **We sincerely appreciate your thoughtful questions, which has helped us better articulate important aspects of our work.** Please let us know if you would like us to provide any additional comments!

---

> > > ### Author Response · Authors · 2025-11-25
> > > **Appreciation for the Review and Willingness to Clarify Further**
> > >
> > > Dear Reviewer hVcV,
> > >
> > > Thank you very much for taking the time to review our work. We greatly appreciate your valuable feedback and suggestions. If there are any remaining questions or if any points in our rebuttal were unclear, please feel free to let us know. We are happy to further clarify and actively participate in the discussion. We will also carefully address the shortcomings in our paper and make improvements in our revised manuscript.
> > >
> > > Best regards,
> > >
> > > The Authors

---

> ### Comment · Reviewer_hVcV · 2025-11-26
>
> Thank you for your clarifications. This paper is now stronger in my view.

---

> > ### Author Response · Authors · 2025-11-26
> >
> > Dear Reviewer hVcV,
> >
> > Thank you very much for the follow-up and for mentioning that **the paper is now stronger in your view.**
> >
> > To ensure that the final assessment accurately reflects the current state of the work, we humbly wonder if you could confirm whether there has been any update to your rating in line with your positive feedback.
> >
> > Thanks again for your review and constructive feedback.
> >
> > Best regards,
> >
> > The Authors

---

### Official Review · Reviewer_gbFN · 2025-10-30

**Soundness:** 3
**Presentation:** 3
**Contribution:** 3
**Rating:** 6
**Confidence:** 2

**Summary:**

This work addresses the problem of catastrophic forgetting in continual learning (CL) models, particularly those using Foundation Models (FMs), which struggle because they rely on simple class comparisons rather than understanding complex objects as compositions of basic concepts.

The proposed solution is CompSLOT, a universal, plug-and-play module that injects concept-level compositionality into any CL method with an FM backbone.

1. It uses a self-supervised Slot Attention mechanism to break down images into low-dimensional representations called slots (concepts).
2. It introduces a primitive selection mechanism to identify and aggregate the most class-relevant concepts from the slots. A primitive loss ensures these primitives are consistent across different examples of the same class.
3.  The core plug-in component is a primitive-logit alignment loss. This loss distills the concept-level similarities between images directly into the model’s final predictions. This regularization guides the model to make decisions based on meaningful, shared, and distinct concepts, rather than simple feature comparisons.

**Strengths:**

Slot attention module is highly stable and shows almost no forgetting across sequential tasks.

CompSLOT is method-agnostic and computationally lightweight as it builds upon the existing FM backbone.

CompSLOT significantly boosts the accuracy of diverse CL baselines.

It enhances compositional generalization abilities.

Benchmarking is broad, and even includes fine-grained classification such as CUB 200.

The work is clearly written, images are readable. Only Figure 1 is too detailed for teaser image and should be simplified to improve the clarity.

Experiments are convincing.

Idea is novel, straightforward and easy to follow.

**Weaknesses:**

The work should be better contextualized in terms of concept-based continual learning, including discussion with work of [1] and follow-up works.

Figure 1 can be improved as it is right now too complex and does not convey the message about novelty well.

Contribution description is vague and unclear. It looks like in the second bullet the CompSlot designed something, not the authors. Looks like the artifact from LLM text improvement. The language there should be simpler, and maybe following organisation made:

- first dot about introduction of compslot and its key components.

- second dot about novel training components as losses

- last one about extensive experiments.

[1] Rymarczyk, Dawid, et al. "Icicle: Interpretable class incremental continual learning." Proceedings of the IEEE/CVF international conference on computer vision. 2023.

**Questions:**

I would like to ask authors for better contextualization of the work and improved clarity.

---

> ### Author Response · Authors · 2025-11-19
> **To Reviewer gbFN**
>
> We sincerely appreciate your constructive comments on this paper. We detail our response below point by point. Please kindly let us know if our response addresses the issues you raised in this paper.
> Please note that the line numbers mentioned below, as well as the section numbers, figure and table numbers, are all based on the original submission. Their relative positions may change in the revision.
>
> ## W1: Improving contextualized in terms of concept-based continual learning
> > - **We appreciate the reviewer for identifying the missing references in the Related Works section**. We would like to clarify our understanding of the mentioned papers and followups:
> >     - ProtoPShare [1], ICICLE[2], and its subsequent extension, LayUP [3] employ **learnable prototypes** to extract concepts from images.
> >     - MuCIL [4] leverages **concepts from the natural language modality**.
> > - We will incorporate the above papers into the *Compositionality subsection* of the *Related Works* in our revised version. Thank you again for your **helpful reminder**.
>
> [1] Rymarczyk, Dawid, et al. "ProtoPShare: Prototypical Parts Sharing for Similarity Discovery in Interpretable Image Classification". KDD 2021
>
> [2] Rymarczyk, Dawid, et al. "Icicle: Interpretable class incremental continual learning." ICCV 2023
>
> [3] Ahrens, K., et al. "Read between the layers: Leveraging multi-layer representations for rehearsal-free continual learning with pre-trained models". TMLR 2024
>
> [4] Susmit Agrawal, et al. "Walking the Web of Concept-Class Relationships in Incrementally Trained Interpretable Models". AAAI 2025
>
>
> ## W2: Figure 1 is too complex and does not convey the message about novelty well
> > - **We sincerely appreciate the reviewer's constructive feedback**. We acknowledge that the current level of detail may obscure our core conceptual contributions.
> > - To improve clarity, we will implement the following revisions:
> >     - **Prominently feature** *Figure 2* (illustrating our main framework) in the *Introduction* section.
> >     - **Enhance the figure caption** with a clear **takeaway** message: "This conceptual pair-wise similarity enables the model to make decisions by additionally considering low-dimensional concept combinations, rather than relying solely on high-dimensional features".
> >     - Relocate *Figure 1* to the *Methods section* where we introduce the slot attention mechanism.
> > - These modifications will help readers better grasp the operational principles of our proposed plug-in approach.
>
>
> ## W3: Contribution description is vague and unclear
> > - **We sincerely appreciate that the current version of the contribution statement may cause misunderstandings.** We will revise the contribution as you suggested as follows:
> >     - We proposed **CompSLOT**, a method-agnostic plug-in comprising:
> >         - a *concept learning module* that leverages *Slot Attention* and *rich vision foundation models* to extract **primitives**, and
> >         - a *concept knowledge distillation module* that enables learners to intentionally discover **shared and distinct concepts** among classes, thereby guiding the decision-making process of classifiers.
> >     - We designed:
> >         - a *primitive selection mechanism* with an additional *primitive loss* that effectively achieves **robust primitive extraction** across different examples of the same class, and
> >         - a *primitive-logit alignment loss* that **contrastively regularizes** the **sample-wise similarities** between continual learners' outputs.
> >     - The experimental results demonstrate that CompSLOT successfully leverages **concept-wise compositionality** to significantly enhance **a wide range** of continual learners.
>
>
>
> **We sincerely appreciate your thoughtful questions, which has helped us better articulate important aspects of our work.** Please let us know if you would like us to provide any additional comments!

---

> > ### Comment · Reviewer_gbFN · 2025-11-19
> > **thanks for the response, can I see the revision?**
> >
> > Thank you for your comments, can you upload the revised version of the manuscript so that I can see the changes? Or is this not possible at this stage?
> >
> > According to the FAQ, it should be possible:
> >
> > For rebuttal revisions, are we limited to one upload or can we update the paper several times?
> >
> > You can upload revisions until the discussion period ends, but reviewers and area chairs are not required to look at every revision. It is up to you to clearly communicate whats been changed.

---

> > > ### Author Response · Authors · 2025-11-20
> > > **Certainly, no problem at all.**
> > >
> > > We apologize for not uploading our revised version when submitting the rebuttal. We have now uploaded the first version of our revision, along with an official comment that highlights the changes we have made.
> > >
> > > Thank you again for your efforts in reviewing our paper.

---

> > > > ### Author Response · Authors · 2025-11-25
> > > > **Appreciation for the Review and Willingness to Clarify Further**
> > > >
> > > > Dear Reviewer gbFN,
> > > >
> > > > We further improve contextualization by describing these works in the Introduction part, see *revision Line 77*.
> > > >
> > > > Thank you very much for taking the time to review our work. We greatly appreciate your valuable feedback and suggestions. If there are any remaining questions or if any points in our rebuttal were unclear, please feel free to let us know. We are happy to further clarify and actively participate in the discussion. We will also carefully address the shortcomings in our paper and make improvements in our revised manuscript.
> > > >
> > > > Best regards,
> > > >
> > > > The Authors

---

> > > > > ### Comment · Reviewer_gbFN · 2025-11-26
> > > > >
> > > > > sorry for late response, I was travelling, so I needed some time to go through revised version. As my concerns were addressed, I raise my score.

---

> > > > > > ### Author Response · Authors · 2025-11-27
> > > > > >
> > > > > > Dear Reviewer gbFN,
> > > > > >
> > > > > > Many thanks for the encouraging feedback and  the positive assessment. We truly appreciate your time and insights.
> > > > > >
> > > > > > Best regards,
> > > > > >
> > > > > > The Authors

---

### Author Response · Authors · 2025-11-20
**Revision notes**

We sincerely thank the reviewers for their valuable efforts in reviewing our paper.
To facilitate tracking of the changes made in the revision, we provide the revision notes as follows:

- **Related Works:** add mentioned prototype-based concept learning methods. (Reviewer gbFN W1)
- **Contributions:** rewrite contributions. (Reviewer gbFN W3)
- **Discussion:** add a paragraph discussing fairness issues in Appendix B. (Reviewer hVcV W1&2)
- **Implementation details:** improve Appendix E to include as many implementation details as possible; highlight backbone information in the Experiment part. (Reviewer hVcV W1&W2&W4 & Reviewer rGQP W1)
- **Influence of hyperparameters:** add a paragraph and Table 3 in the experiment section which showcase an example of hyperparameter influences in Appendix H. (Reviewer hVcV W3 & Reviewer rGQP W3)
- **Experiment on Finetuning:** add a paragraph in Section M and Table 12 which showcase the results on Finetuning. (Reviewer rGQP Q2)
- **Concept visualization:** add a paragraph and Figure 4 in the experiment section which summarizes the observations in Appendix K and Appendix H. (Reviewer rGQP W2)
- **Future work:** add a future work. (Reviewer hVcV W3)
- **Figure rearrangement:** Put the framework figure (originally Figure 2) to Introduction part and add a Takeaway message. (Reviewer gbFN W2 & Reviewer hVcV W3)
- **Figure rearrangement:** Put the learned slot example figure (originally Figure 1) to Method part and describe the necessary messages in the main content. (Reviewer gbFN W2)

---

### Author Response · Authors · 2025-12-03
**Summary of the discussion phase (1/2)**

Dear Area Chairs,

We sincerely thank you for your time and effort in overseeing the review process, especially under this year's added complexity due to the OpenReview identity‐leakage incident. We truly understand that navigating such uncertainty while ensuring fairness and quality places significant demands on your role.

In light of this, and following the program chairs' suggestion, we have provided a concise and transparent summary of the discussion-phase interactions to serve as a reference that eases your decision making.

Our paper
> - proposes a **plug-in module**, CompSLOT, designed to improve off-the-shelf continual learning algorithms by equipping them with greater **compositionality**.
>    - It employs unsupervised Slot Attention to **extract latent concepts** from images, followed by a learnable selection mechanism that identifies **class-relevant primitives**.
>    - It then distills **pairwise primitive similarities** across batched samples into the logits via a contrastive loss.
>    - These concept-level signals encourage the model to classify based on **combinations of class-relevant concepts** in a compact, low-dimensional space. The composition of these concepts further supports rapid adaptation to new tasks.
> - empirically demonstrates that CompSLOT effectively leverages **concept-wise compositionality**, yielding significant improvements for eight continual learning algorithms across (1) compositional benchmarks, (2) coarse-grained CL benchmarks, and (3) fine-grained CL benchmarks.

The positive acknowledgments from reviewers include
> - **Paper Quality and Clarity:** Reviewers consistently praise the clarity and readability of the paper, noting it is **well-written and easy to follow** (Reviewers `rGQP`, `gbFN`). A minor comment about **Figure 1 being overly detailed** (Reviewer `gbFN`) has already been addressed in the revision.
> - **Novelty:** Reviewers agree that the paper presents a **novel and meaningful conceptual direction** (Reviewers `gbFN`, `rGQP`). They also emphasize the method’s ability to extract **disentangled, class-relevant concepts** (Reviewer `hVcV`), reinforcing its conceptual significance.
> - **Generalizability:** The design of CompSLOT is regarded as **stable, lightweight, and method-agnostic** (Reviewers `gbFN`, `hVcV`). They further note that the method can be **easily integrated into diverse continual learning frameworks** (Reviewer `rGQP`).
> - **Empirical Validation:** Reviewers unanimously commend the **comprehensive and convincing** empirical evaluation, noting that CompSLOT **consistently boosts performance** across a wide range of continual learning baselines and benchmarks (Reviewers `gbFN`, `hVcV`, `rGQP`).

In the discussion period, we **responded to all weaknesses and questions** raised by the three reviewers and **updated the revision pdf file accordingly** with all changes highlighted in blue.

We are encouraged that prior to the large-scale identity leakage, all three reviewers explicitly acknowledged that their concerns had been addressed, resulting in updated scores of `8, 6, 6` with positive average `6.67`.
> - **Reviewer `gbFN`** **raised his/her score from 6 to 8 on Nov 26 18:45 UTC[(OpenReview link)](https://openreview.net/revisions?id=U6heZiOYnW)** (one day before the large-scale identity leakage happened).
> - **Reviewer `rGQP`** **raised his/her score from 4 to 6 on Nov 26 15:52 UTC[(OpenReview link)](https://openreview.net/revisions?id=NRAnJWTEMg)** (one day before the large-scale identity leakage happened).
> - **Reviewer `hVcV`** commented “This paper is now stronger in my view” and maintained his/her rating on Nov 26 3:05 UTC[(OpenReview link)](https://openreview.net/revisions?id=uGP33QOMtz) after we gave clarifications([OpenReview link1](https://openreview.net/revisions?id=rKHOpCf8Ow),[link2](https://openreview.net/revisions?id=Wp0Z6u68O6),[link3](https://openreview.net/revisions?id=q2cehnj1Zw)).

---

> ### Author Response · Authors · 2025-12-03
> **Summary of the discussion phase (2/2)**
>
> In the below thread, we provide a more detailed breakdown of the reviewer discussions to further clarify the points addressed and the consensus reached.
>
> **Reviewer `gbFN` (The initial score is 6 and raised to 8 on Nov 26 18:45 UTC)**
> > - **W1:** We need to *improve contextualization* in terms of concept-based continual learning.
> >   - We incorporated the missing papers into the **“Compositionality”** subsection and referenced them in the **Introduction** in our **new revision**.
> > - **W2:** *Figure 1* was too complex to convey the message about novelty.
> >   - We moved *Figure 2*, which highlighted our main mechanisms, to the **Introduction** section and added a *takeaway* message.
> >   - We relocated *Figure 1* to the **Methods** section after introducing the Problem Definition.
> > - **W3:** We need to *improve Contributions* as the section is vague and unclear.
> >   - We completely revised the **contribution** statement to be precise about what “CompSLOT” does.
> > -----------------
> >  **Outcome:** Reviewer `gbFN` **said “As my concerns were addressed, I raise my score.”[(OpenReview link)](https://openreview.net/revisions?id=U6heZiOYnW)**
>
>  **Reviewer `hVcV` (The initial score is 6)**
> > - **W1&W2:** We need to clearly specify the efforts we made to achieve *fairness*.
> >     - We complemented detailed implementation settings and specifically a *discussion section* in **Appendix B** in our **new revision**, and highlighted the ablation study we conducted to show that performance gains are not from the increased model capacity.
> > - **W3:** How does CompSLOT work with baseline methods and would it conflict CPrompt's idea of constraining logits?
> >     - We highlighted our main machanisms and why they worked with the existing supportive experimental results in **Table 1** and visual exploration in **Appendix K**.
> >     - We *conducted compatibility test experiments* varying the  coefficient of the primitive-logit alignment loss $\beta$, showing that the method works with CPrompt within a reasonable range.
> >     - We incorporated the observations in one paragraph in the **experimental part** of our **new revision**.
> > - **W4:** The method is too *complex* to reproduce easily.
> >     - We provided a Python pseudocode in the rebuttal to demonstrate that the implementation of our plugin is lightweight and promised to release the full code.
> > ------------------------
> > **Outcome:** The reviewer acknowledged our clarifications and said “This paper is now stronger in my view”[(OpenReview link)](https://openreview.net/revisions?id=uGP33QOMtz).
>
> **Reviewer `rGQP` (The initial score is 4 and raised to 6 on Nov 26 15:52 UTC)**
> > - **W1:** We need to specify detailed experimental settings.
> >   - We provided detailed implementation settings and a *discussion section* in **Appendix B** in our **new revision**.
> > - **W2&W3:** lacking concept analysis and hyperparameter studies.
> >   -  We pointed out that the concept analysis is in **Appendix K** and hyperparameter studies are in **Appendix H**.
> >   - We also provided a brief summary regarding visualization and a discussion about hyperparameters in the **experimental part** in our **new revision**.
> > - **Q1:** Whether is our CompSLOT suitable on other backbones?
> >    - We pointed out that we had studied *different architectures of vision models* and *different pretraining strategies* in **Appendix J**, which justified that our CompSLOT is **generalizable to any ViT**.
> > - **Q2:** We need to conduct experiments on the naive continual learner: finetuning.
> >   - We conducted the experiments on finetuning where only the classifier is extendable and trained while the feature extractor is frozen.
> >   - We observed that CompSLOT successfully acieves higher *average task accuracy* with smaller *average forgetting*.
> >   - We updated the results and discussion in **Section M** and **Table 12** in our **new revision**.
> >--------------------
> >**Outcome:** Reviewer `rGQP` responded with **“Thank you for your response. Most of my concerns have been addressed... Therefore, I have increased my score”[(OpenReview link)](https://openreview.net/revisions?id=NRAnJWTEMg) from 4 to 6**.
>
> Again, we sincerely appreciate your time handling our submission and look forward to the possibility of our work contributing meaningfully to the field.
>
> Best Regards,
>
> Authors

---

### Meta-Review · Area_Chair_mfjn · 2025-12-07

**Summary:**

This paper makes a strong and original contribution to the field. It is clearly written and easy to follow, and its experimental results are thorough and well-supported.

The authors have also addressed all reviewer feedback in detail, and all reviewers now agree on acceptance. I therefore recommend accepting this paper.

**Reviewer Concerns:**

All concerns have been well addressed.

**Reviewer Scores:**

Original scores: 6,4,6

According to the reviewer-author discussion records,

Reviewer gbFN and rGQP agreed to raise the score.
Reviewer hVcV commented “This paper is now stronger in my view” and maintained the rating.

---

### Decision · Program_Chairs · 2026-01-26

Accept (Oral)